# Towards Certification of Uncertainty Calibration under Adversarial Attacks

**Cornelius Emde**[1]*, **Francesco Pinto**[1], **Thomas Lukasiewicz**[2,1], **Philip Torr**[1], **Adel Bibi**[1]
[1]University of Oxford    [2]Vienna University of Technology

## Abstract

Since neural classifiers are known to be sensitive to adversarial perturbations that alter their accuracy, *certification methods* have been developed to provide provable guarantees on the insensitivity of their predictions to such perturbations. Furthermore, in safety-critical applications, the frequentist interpretation of the confidence of a classifier (also known as model calibration) can be of utmost importance. This property can be measured via the Brier score or the expected calibration error. We show that attacks can significantly harm calibration, and thus propose certified calibration as worst-case bounds on calibration under adversarial perturbations. Specifically, we produce analytic bounds for the Brier score and approximate bounds via the solution of a mixed-integer program on the expected calibration error. Finally, we propose novel calibration attacks and demonstrate how they can improve model calibration through *adversarial calibration training*.

## 1 Introduction

The black-box nature of deep neural networks and the unreliability of their predictions under several forms of data shift hinder their deployment in safety-critical applications (Abdar et al., 2021; Linardatos et al., 2021). It is well known that neural network classifiers are extremely sensitive to perturbations $\gamma$ on image $x$ that are small enough to be imperceptible to the human eye, yet $x + \gamma$ yields a different prediction than $x$ (Goodfellow et al., 2015; Szegedy et al., 2014; 2016). While a variety of methods have been proposed to improve the *empirical* robustness of neural networks, *certification methods* have recently gained traction, since they provide *provable guarantees* on the invari-

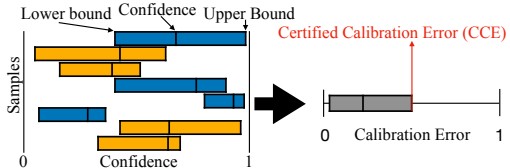

Figure 1: This work proposes a certificate (upper bound) on the calibration error of a classifier under adversaries. Each box on the left represents one prediction from a *certified model*: The range of each box represents a certificate on the confidence score under adversaries, the color whether the prediction is certifiably correct. We translate these certificates into a certificate on the calibration error (*right*), i.e. a worst-case under attack.

ance of predictions under adversarial attacks (Wong and Kolter, 2018) (see Appendix A.1 for a detailed introduction). Further, existing work provides bounds on the confidence scores (i.e., the top value of the softmax output vector) of certified models under adversarial inputs (Kumar et al., 2020). However, they do not inform us about the average mismatch between accuracy and confidence. The reliability of a classifier in these regards is generally quantified by the *expected calibration error* (ECE) (Naeini et al., 2015) and the *Brier score* (BS) (Brier, 1950; Bröcker, 2009). Both describe the calibration of confidences across a set of predictions and play a strategic role in applications that leverage confidence scores in their decision-making process. For instance, one might utilize reliable confidence scores in medical diagnostics to decide whether to trust the machine learning classifier without further human intervention or whether to defer the decision to a human expert. However, we empirically show that it is possible to produce adversaries that severely impact the reliability of confidence scores while leaving the accuracy unchanged, even when the classifier is explicitly trained to be robust to adversarial attacks on the accuracy.

---

*Corresponding Author `cornelius.emde@cs.ox.ac.uk`.

For this reason, we propose the notion of *certified calibration* to provide guarantees on the calibration of certified models under adversarial attacks. While certification of individual predictions enables guarantees on the accuracy of a classifier under adversaries, the additional certificates on the confidences enable worst-case bounds on the *reliability* of confidence scores.

**Our contributions are the following:**

- We identify and demonstrate attacks on model calibration that severely impact uncertainty estimation while remaining unnoticed when solely monitoring model accuracy.
- We introduce *certified calibration* quantified through the *certified Brier score* (CBS) and the *certified calibration error* (CCE). For the former, we present a closed-form bound as certificate, while we show a *mixed-integer non-linear program* reformulation for the latter with an effective solver, dubbed *approximate certified calibration error* (ACCE).
- We introduce a new form of adversarial training, *adversarial calibration training* (ACT), that augments training with Brier and ACCE adversaries. We obtain ACCE adversaries by solving a *mixed-integer program* and demonstrate that ACT can improve certified uncertainty calibration.

## 2 CONFIDENCE CALIBRATION

We first formally introduce the notion of calibration in Section 2.1; then, in Section 2.2, we show how attacks on confidence scores and calibration metrics may be beneficial to an attacker. Finally, in Section 2.2, we show that such attacks are not only theoretically possible but also realizable.

### 2.1 QUANTIFYING THE CONFIDENCE-ACCURACY MISMATCH

Let $\mathcal{D} = \{\mathbf{x}_n, y_n\}_{n=1}^N$ be a dataset of size $N$ with $\mathbf{x}_n \in \mathbb{R}^D$, $y_n \in \mathcal{Y} = \{1, 2, \ldots, K\}$, and $f : \mathbb{R}^D \to \Delta_K$ be a neural network, where $\Delta_K$ is a probability simplex over $K$ classes. We denote the $k$-th component of $f$ as $f_k$ and define $F : \mathbb{R}^D \to \mathcal{Y}$ to be the *hard classifier* predicting a class label obtained by $F(x) := \arg\max_{k \in \mathcal{Y}} f_k(x)$. This prediction is done with the *confidence* provided by the *soft classifier* $z : \mathbb{R}^D \to [0, 1]$, obtained through $z(x) := \max_{k \in \mathcal{Y}} f_k(x)$. With slight abuse of notation, we refer to functions and their outputs simultaneously, e.g., $z \in [0, 1]$ is an output of $z(x)$. The notion of reliability of uncertainty estimates is typically quantified through two different but related quantities that we now introduce: the *expected calibration error* and the *Brier score*.

**Expected Calibration Error.** For classification tasks, calibration describes a match between the model's confidence and its empirical performance (DeGroot and Fienberg, 1983; Naeini et al., 2015). A well-calibrated model predicts with confidence $z$ when the fraction of correct predictions is exactly $z$, i.e., $\mathbb{P}(F = y | Z = z) = z$. This enables us to interpret $z$ as a probability in the frequentist sense. The expected calibration error is the expected difference between confidence and accuracy over the distribution of confidence $Z$, i.e., $\text{ECE} = \mathbb{E}_Z \left[ |\mathbb{P}(F = y | Z = z) - z| \right]$. A common estimator is obtained by discretizing the empirical distribution over $Z$ through binning (Guo et al., 2017). For each set of confidences $B_m$ in the $m$-th bin, the average confidence is compared with the accuracy:

$$\hat{\text{ECE}} = \sum_{m=1}^M \frac{|B_m|}{N} \left| \frac{1}{|B_m|} \sum_{n \in B_m} c_n - \frac{1}{|B_m|} \sum_{n \in B_m} z_n \right|, \tag{1}$$

where $N$ is the number of samples, $z_n$ the confidence of the $n$-th prediction $F_n$, and $c_n$ is the prediction correctness, i.e., $c_n = 1$ if $F_n = y_n$, and 0 otherwise. Multiple variants of the calibration error and its estimators exist (Kumar et al., 2019). As commonly done in the literature, we focus on top label calibration that ignores the calibration of confidences of lower-ranked predictions. When using equal-width interval binning for the estimator in (1), we refer to it as ECE, and when using an equal-count binning scheme, we refer to it as AdaECE (Nguyen and O'Connor, 2015; Nixon et al., 2019).

**Brier Score.** Accuracy and calibration represent different concepts, and one may not infer one from the other unambiguously (see Appendix B.1). The two concepts are unified under *proper scoring rules*, such as the *Brier Score* (BS) (Brier, 1950), which is commonly

used in the calibration literature. It has been shown that this family of metrics can be decomposed into a calibration and a refinement term (Murphy, 1972; 1973; Bröcker, 2009). An optimal score can only be achieved by predicting accurately and with appropriate confidence. The BS is mathematically defined as the mean squared error between the confidence vector $f$ and a one-hot encoded label vector. Here, we focus on the *top-label Brier score* TLBS $= N^{-1}\|\mathbf{c} - \mathbf{z}\|_2^2$, which is the mean squared error between the $N$ confidences $\mathbf{z} \in [0,1]^N$ and correctness of each prediction $\mathbf{c} \in \{0,1\}^N$.

## 2.2 Calibration under Attack

**Motivation.** A wide range of adversarial attacks has been discussed in the literature, with label attacks being the most prevalent (Akhtar and Mian, 2018). However, their immediate effects on the reliability of uncertainty estimation are often overlooked: Label attacks do not only impact the correctness of predictions, but also the confidence scores, both affecting uncertainty calibration. Beyond label attacks, techniques directly targeting confidence scores have been developed. Galil and El-Yaniv (2021) show that such attacks can cause significant harm in applications where confidence scores are used in decision processes at a sample level, e.g., for risk estimation and credit pricing in credit lending. Yet, they evade detection by leaving predicted labels unchanged. We note that this impacts system-level uncertainty and conclude that both label and confidence attacks leave system calibration vulnerable.

Furthermore, machine learning systems incorporating uncertainty estimates in their decision process are monitored regularly for both their *predictive performance* and *calibration*, particularly in safety-critical applications. When abnormalities in either monitored metric arise, the model may be pulled from deployment for further investigation, resulting in major operational costs. To this end, an attacker might produce attacks on the accuracy of a model to induce a *Denial-of-Service* (DoS). While defenses against label attacks effectively protect the model accuracy, the *calibration* is not sufficiently protected: An attacker can directly target system calibration, causing harm due to increased operational risk, and cause changes in monitored calibration metrics, thus forcing a DoS. Despite extensive countermeasures to adversarial attacks, no existing technique addresses this vulnerability.

Extending the credit lending example, deploying a certified model provides provable protections: First, the accuracy of predictions on customer defaults is protected because of the provable invariance of predicted labels under attack (Cohen et al., 2019). Second, the uncertainty per individual customer is protected by bounds on the confidence scores (Kumar et al., 2020). However, these protections do not inform the operational risk of the system, and the reliability of uncertainty estimation across the system remains vulnerable. Miscalibrated uncertainty can lead to underestimation of risks, resulting in unexpected customer defaults, or overestimation, rendering credit pricing uncompetitive and reducing revenue. It is, therefore, essential for the credit lender to monitor calibration rigorously. An attacker might exploit this by directly targeting calibration with severe consequences for the credit lender.

**Feasibility of Calibration Attacks.** After discussing possible reasons for attacks on calibration, we now demonstrate that these are feasible. Galil and El-Yaniv (2021) show that it is possible to produce attacks that leave the accuracy unchanged while degrading the Brier score. Expanding on their Attacks on Confidence Estimation (ACE), we introduce a family of parameterised ACE attacks on the cross-entropy loss $\mathcal{L}_{CE}$ that we call $(\eta, \omega)$-ACE attacks. These attacks solve the following objective:

$$\max_{\|\gamma\|_p \leq \epsilon, F(x+\gamma)=F(x)} \eta \mathcal{L}_{CE}(f(x+\gamma), \omega), \quad (2)$$

where $\eta \in \{1, -1\}$, $\omega \in \{y, \hat{y}\}$, $\hat{y} \triangleq F(x+\gamma)$ is the prediction and $y$ the true label. For

Table 1: AdaECE↓ (%) when performing our $(\eta, \omega)$-ACE attacks compared to the case in which the attack is not performed for PreAct-ResNet18 and ResNet50 on CIFAR-10 and ImageNet for models fitted through Standard Training (ST) and Adversarial Training (AT).

| | | | CIFAR 10 | ImageNet | |
|---|---|---|---|---|---|
| | Unattacked: | | 3.06 | 3.70 | |
| | $\eta$ | $\omega$ | $\epsilon = 8/255$ | $\epsilon = 2/255$ | $\epsilon = 3/255$ |
| ST | $-1$ | $y$ | 2.49 | 1.06 | 10.62 |
| | $+1$ | $y$ | 18.79 | 47.23 | 46.92 |
| | $-1$ | $\hat{y}$ | 5.19 | 23.72 | 23.73 |
| | $+1$ | $\hat{y}$ | 11.87 | 25.17 | 27.84 |
| | Unattacked: | | 20.69 | 9.03 | |
| | $\eta$ | $\omega$ | $\epsilon = 8/255$ | $\epsilon = 2/255$ | $\epsilon = 3/255$ |
| AT | $-1$ | $y$ | 21.84 | 7.36 | 8.51 |
| | $+1$ | $y$ | 23.51 | 11.62 | 12.21 |
| | $-1$ | $\hat{y}$ | 11.92 | 0.91 | 3.42 |
| | $+1$ | $\hat{y}$ | 25.59 | 13.54 | 14.28 |

$\eta = 1$, this attack decreases model confidence in label $y$ or prediction $\hat{y}$, and for $\eta = -1$, it increases confidence, both without altering the label (i.e., $F(x + \gamma) = F(x)$). In Table 1, we

show that all four possible configurations of our $(\eta, \omega)$-ACE can be effective at significantly altering the ECE of PreActResNet18 (He et al., 2016a) and ResNet50 (He et al., 2016b) on the validation set of CIFAR-10 (Krizhevsky, 2009) and ImageNet-1K (Deng et al., 2009), respectively. The attacks can even be effective on robustly trained models. For a ResNet50 on ImageNet, a $(1, y)$-ACE attack with radius $\epsilon = 2/255$ can increase the ECE from 3.70 to 47.23. Performing adversarial training can partly alleviate this issue, but a $(1, \hat{y})$-ACE attack can still increase the ECE from 9.03 to 13.54. This clearly indicates that it is possible to significantly manipulate the calibration of a model while preserving its accuracy. In Appendix B.2, we provide reliability diagrams that illustrate the effect of each $(\eta, \omega)$-ACE attack in addition to more details on the experimental setup.

## 3 Certifying Calibration

We propose to use *certification* as a framework to protect calibration under adversarial attacks. Hence, in this section, we first introduce existing methods to certify predictions and then build upon them to introduce *certified calibration*.

**Prerequisites.** Certification methods establish mathematical guarantees on the behavior of model predictions under adversarial perturbations. Specifically, for a given input $\mathbf{x}_n$, the certified model issues a set of guarantees (such as the invariance of predictions) for all perturbations $\boldsymbol{\gamma}_n$ up to an adversarial budget of $R_n$ in some norm $\| \cdot \|$, i.e., $\|\boldsymbol{\gamma}_n\| \le R_n$. Practitioners deploying a system with a designated threat model of attacking budget $R$ receive a guarantee on each prediction with $R_n \ge R$. This enables an estimate of the model's performance under adversaries. Most commonly discussed is the *certified accuracy* at $R$, the fraction of certifiably correct predictions. Our method assumes two certificates.

**Assumption 3.1.** Model predictions are invariant under all perturbations,

$$\forall \|\boldsymbol{\gamma}_n\| \le R_n : \quad F(\mathbf{x}_n + \boldsymbol{\gamma}_n) = F(\mathbf{x}_n). \tag{C1}$$

**Assumption 3.2.** Prediction confidences are bounded under perturbations,

$$\forall \|\boldsymbol{\gamma}_n\| \le R_n : \quad l\left(z(\mathbf{x}_n), R_n\right) \le z(\mathbf{x}_n + \boldsymbol{\gamma}_n) \le u(z(\mathbf{x}_n), R_n). \tag{C2}$$

We further, assume that $R_n$, $l(\cdot)$ and $u(\cdot)$ can be computed for each sample $\mathbf{x}_n$. A state-of-the-art method to obtain such $\ell_2$ certificates on large models and datasets is *Gaussian smoothing* (Cohen et al., 2019). A smooth classifier is constructed from a base classifier by augmenting model inputs with Gaussian perturbations, $\boldsymbol{\delta} \sim N(0, \sigma^2 \mathbf{I}_D)$, and aggregating predictions. We refer to $\bar{F} : \mathbb{R}^D \to \mathcal{Y}$ as the *smoothed hard classifier*, returning the predicted class and providing radius $R_n$, for which the prediction is invariant (see Certificate **C1**). The certificate on the prediction has been extended by Salman et al. (2019) and Kumar et al. (2020), showing a certificate on the confidence. We denote the smooth model returning a confidence vector as $\bar{f} : \mathcal{X} \to \Delta_K$ and denote the *smoothed soft classifier* as $\bar{z} : \mathbb{R}^D \to [0, 1]$, obtained through $\bar{z}(\mathbf{x}) = \max_{k \in \mathcal{Y}} \bar{f}_k(\mathbf{x} + \boldsymbol{\delta})$. $\bar{z}$ indicates the confidence of prediction $\bar{F}$. Kumar et al. (2020) provide two different bounds on $\bar{z}$ as certificate **C2**, the STANDARD and the CDF certificate. In Appendix A.1, we provide more details on these, as well as a general introduction to Gaussian smoothing.

### 3.1 Certifying Brier Score

As discussed earlier, the Brier score is a unified measure of a model's discriminative performance and calibration, and is widely used in the literature on model calibration. We introduce the *certified Brier score* (CBS) as the worst-case (i.e. largest) top label Brier score under any perturbation $\|\boldsymbol{\gamma}_n\| \le R$ over a set of $N$ samples. We assume a model returning certified predictions and providing lower and upper bounds $\mathbf{l}, \mathbf{u} \in \mathbb{R}^N$ on the top confidence $\mathbf{z} \in \mathbb{R}^N$ given the certified radius $R$ on the input perturbation, i.e. certificate **C2**. Based on this, we state the following upper bound on the Brier score.

**Theorem 3.3.** *Let $\mathbf{l}$, $\mathbf{u}$ be the bounds on $\mathbf{z}$, and $\mathbf{z}$ be the output of a certified classifier. Further, let $\mathbf{c} \in \mathbb{R}^N$ be the indicator that predictions are correct. The upper bound on the Brier score is given by:*

$$\max_{\mathbf{l} \le \mathbf{z} \le \mathbf{u}} TLBS(\mathbf{z}, \mathbf{c}) = \frac{1}{N} \| \mathbf{c} - \mathbf{l} \, \mathbf{c} - \mathbf{u} \, (\mathbf{1} - \mathbf{c}) \|_2^2, \tag{3}$$

*where the products are element-wise. Refer to Appendix D for the proof.*

This bound is tight and relies on the fact that shifting the confidences leaves each certified prediction $F$ and thus $\mathbf{c}$ unchanged. Therefore, an adversary cannot flip the prediction to increase the *confidence gap*, the sample-level distance between correctness $c_n$ and confidence $z_n$. The Brier score is maximized when the confidence gap is large, the opposite of a good classifier's expected behavior.

## 3.2 CERTIFYING CALIBRATION ERROR

While both the Brier score and the expected calibration error capture some notion of calibration, the confidence scores bounding the Brier score (3) do not necessarily bound the ECE. We show that the ECE can be increased even further. Therefore, we introduce a definition for the *certified calibration error* and provide a method to approximate it.

**Definition 3.4.** The *certified calibration error* (CCE) on a dataset $\mathcal{D} = \{\mathbf{x}_n, y_n\}_{n=1}^N$ at radius $R$ is defined as the maximum ECE that can be observed as a result of perturbations on the input within an $\ell_2$ ball of radius $R$. Let, $\boldsymbol{\gamma}_n$ be the perturbation on input $\mathbf{x}_n$. Thus, the CCE is:

$$\text{CCE} = \max_{\forall n: \|\boldsymbol{\gamma}_n\|_2 \leq R} \hat{\text{ECE}}\left([\bar{z}(\mathbf{x}_n + \boldsymbol{\gamma}_n)]_{n=1}^N, \mathbf{y}\right). \tag{4}$$

This is the largest estimated calibration error that is possible on a dataset if every sample is perturbed by at most $R$. Such a bound on the calibration is fundamentally different from a bound on the confidence: A certificate on the confidence as described in **C2** bounds the confidence *per sample*. The CCE in (4) bounds the error between confidence and frequentist probability *across a set of samples*. Finding the bound in (4) is not trivial, as the estimator of the ECE in (1) is neither convex nor differentiable. Therefore, we propose a numerical method to estimate (4) and provide an empirical, approximate certificate, the *approximate certified calibration error* (ACCE).

## 3.3 CCE AS MIXED-INTEGER PROGRAM

We show in this section that we may optimize (4) by reformulating the problem into a mixed-integer program (MIP). In the following, we first introduce the intuition and notation to rewrite the objective and state equivalence of (5) to (4) in Theorem 3.5. Subsequently, the constraints for equivalence are introduced in detail. We conclude by restating the problem in canonical form for clarity. Readers unfamiliar with MIP may refer to Appendix A.3 for a brief introduction. In Appendix E.2, we provide a numerical example to illustrate the derivations presented in this section.

While the problem in (4) is defined over perturbations $\|\boldsymbol{\gamma}_n\|_2 \leq R$ for each input $\mathbf{x}_n$, we rely on certificate **C2** and directly optimize over the confidences. Since the ECE estimator in (1) uses bins to estimate average confidence and accuracy, maximizing the estimator requires solving a bin assignment problem, where $N$ confidence scores $z_n$ are assigned to $M$ bins. We show how to jointly solve this assignment problem and find the values of $z_n$ maximizing the calibration estimator across bins.

When perturbing the input $\mathbf{x} + \gamma$, the bin assignment might change: $\bar{z}(\mathbf{x})$ might belong to bin $m$, but $\bar{z}(\mathbf{x} + \gamma)$ to $m' \neq m$.[1] While the assignment is determined by the confidence score, it is key to our reformulation to split these into separate variables. The motivation is that very small changes in $z_n$ might lead to a shift in bin assignment and thus contribute differently to the calibration error. We define the binary assignment $a_{n,m}$ of confidence $z_n$ to bin $m$ and accordingly define $\mathbf{a} = [a_{1,1}, \ldots, a_{1,M}, a_{2,1}, \ldots, a_{N,M}]^\top \in \{0,1\}^{NM}$, where $N$ is the number of samples and $M$ the number of bins. The confidence scores maximizing the calibration error might be different across bins, i.e., it is possible that the worst-case for bin $m$ is different than for bin $m' \neq m$: $z_{n,m}^* \neq z_{n,m'}^*$. Therefore, we model the confidence independently for each bin, introducing bin-specific confidences $z_{n,m}$ and with it $\mathbf{z} = [z_{1,1}, \ldots, z_{1,M}, z_{2,1}, \ldots, z_{N,M}]^\top \in [0,1]^{NM}$. Further, let $c_n$ be the indicator whether prediction $n$ is correct, i.e., $c_n = \mathbb{I}\{\bar{F}(\mathbf{x}_n) = y_n\}$ and let $e_{n,m} = c_n - z_{n,m}$, the sample confidence gap. Note that $c_n$ is independent of the bin assignment as a result of certification, i.e., while the confidence may shift, the prediction will remain unchanged. Analog to $\mathbf{a}$ and $\mathbf{z}$, we define $\mathbf{e} = [e_{1,1}, \ldots, e_{1,M}, e_{2,1}, \ldots, e_{N,M}]^\top \in \mathbb{R}^{NM}$. Now let $\mathbf{B}$ be a stack of $N$ identity

---

[1] In this section, we relax the notation of $\bar{z}$ to $z$, as we are assuming smoothed models everywhere.

matrices of size $M$, i.e., $\mathbf{B} = [\mathbf{I}_M, \ldots, \mathbf{I}_M]^\top$. We define $\mathbf{E}(\mathbf{z}) = (\mathbf{e}\mathbf{1}_M^\top) \odot \mathbf{B} \in \mathbb{R}^{NM \times M}$ where $\odot$ is the Hadamard product and $\mathbf{1}_M$ is the 1-vector of size $M$. We can now rewrite the calibration error.

**Theorem 3.5.** *Let $\mathbf{a}$ and $\mathbf{E}$ be as defined above, and $\mathbf{z}$ be the output of a certified classifier. The calibration error estimator in (1) can be expressed as:*

$$\hat{ECE} = \frac{1}{N}\|\mathbf{E}(\mathbf{z})^\top \mathbf{a}\|_1. \tag{5}$$

*Maximizing (5) over $(\mathbf{a}, \mathbf{z})$ is equivalent to solving (4) when subjecting $\mathbf{a}$ and $\mathbf{z}$ to the Unique Assignment Constraint, Confidence Constraint, and Valid Assignment Constraint below. Refer to Appendix E.1 for a proof.*

**Unique Assignment Constraint.** The assignment variable $\mathbf{a}$ has to be constrained such that each data point is assigned to exactly one bin, i.e., $\sum_m a_{n,m} = 1 \ \forall n$. To this end, we define $\mathbf{C} = \mathbf{I}_N \otimes \mathbf{1}_M \in \mathbb{R}^{NM \times N}$, where $\otimes$ is the Kronecker product. $\mathbf{C}$ sums up all assignments per data point, and hence the constraint is $\mathbf{C}^\top \mathbf{a} = \mathbf{1}_N$.

**Confidence Constraint.** Let $l_n^z \leq z_{n,m} \leq u_n^z$ be the lower and upper bound on the confidence as provided by the certificate **C2**. In addition, any confidence assigned to bin $m$ has to adhere to the boundaries of this bin, i.e., $l_m^B \leq z_{n,m} \leq u_m^B$. We can combine these two conditions to unify the bounds: $\max(l_n^z, l_m^B) = l_{n,m} \leq z_{n,m} \leq u_{n,m} = \min(u_n^z, u_m^B)$. With this, we define $\mathbf{l} = [l_{1,1}, \ldots, l_{1,M}, l_{2,1}, \ldots, l_{N,M}]^\top$ and $\mathbf{u} = [u_{1,1}, \ldots, u_{1,M}, u_{2,1}, \ldots, u_{N,M}]^\top$ and state the full constraint: $\mathbf{l} \leq \mathbf{z} \leq \mathbf{u}$.[2]

The bounds on the confidence per bin may not intersect with the certificates on the confidence, i.e., for some $n, m$ $[l_n^z, u_n^z) \cap [l_m^B, u_m^B) = \emptyset$. This is expected for narrow certificates on $z$ or a large number of bins. For these instances, we set $\mathbf{z}$ to 0 and define $\mathbf{l}'$ and $\mathbf{u}'$ to be $\mathbf{l}$ and $\mathbf{u}$, respectively, with the same elements set to 0. Thus, the feasible constraint set on the confidence per bin is defined as $S_z = \{\mathbf{z} : \forall n, m : l'_{n,m} \leq z_{n,m} \leq u'_{n,m}\}$.

**Valid Assignment Constraint.** We have identified that some confidences can never be assigned to some bins when the confidence certificates and bin boundaries do not intersect. For those instances, we constrain $a_{n,m} = 0$. Let $k_{n,m} = \mathbb{I}\{l_{n,m} \geq u_{n,m}\}$ be the indicator that bin $m$ is inaccessible to data point $n$. We define matrix $\mathbf{K}$ to be a $NM \times N$ matrix summing all inaccessible bin assignments. Formally, letting $\mathbf{k} = [k_{1,1}, \ldots, k_{1,M}, k_{2,1}, \ldots, k_{N,M}]^\top$, and $\mathbf{K} = \mathbf{k}\mathbf{1}_N^\top \odot (\mathbf{I}_N \otimes \mathbf{1}_M)$, the constraint is: $\mathbf{K}^\top \mathbf{a} = \mathbf{0}_N$.

**Formal Program Statement.** We summarize the constraints above and restate the program introduced in Theorem 3.5 in its canonical form for clarity. The MIP over $(\mathbf{a}, \mathbf{z})$ is given by:

$$\text{maximize} \quad \frac{1}{N}\|\mathbf{E}(\mathbf{z})^\top \mathbf{a}\|_1 \quad \text{subject to:} \quad \mathbf{a} \in \{0,1\}^{NM}, \mathbf{C}^\top \mathbf{a} = \mathbf{1}_N, \mathbf{K}^\top \mathbf{a} = \mathbf{0}_N, \mathbf{z} \in S_z. \tag{6}$$

This problem is sparse in objectives and constraints as $\mathbf{C}$, $\mathbf{K}$ and $\mathbf{E}$ contain at most $NM$ non-zero elements. The reformulation of (4) into (6) provides us with a useful framework to run a numerical solver.

## 3.4 ADMM Solver

We propose to use the ADMM algorithm (Boyd et al., 2011) to solve (6). While ADMM has proofs for convergence on convex problems, it enjoys good convergence properties even on non-convex problems (Wang et al., 2019). ADMM minimizes the augmented Lagrangian of the constrained problem by sequentially solving sub-problems, alternating between minimizing the primal variables and maximizing the dual variables. We introduce split variables to show that the problem can be reformulated into the standard ADMM objective (see Appendix E for more details). Due to the sparsity of the problem, ADMM has linear complexity in the number of bins and samples, $\mathcal{O}(NM)$. In our experiments, ADMM always converges in

---

[2]In Section 3.1 on the CBS, $\mathbf{l}, \mathbf{u} \in [0,1]^N$ are the immediate bounds on $\mathbf{z} \in [0,1]^N$ provided by the certificate on the confidences. Here, $\mathbf{l}, \mathbf{u} \in [0,1]^{NM}$ provide bounds on the expanded $\mathbf{z} \in \mathbb{R}^{NM}$ as defined in this section, now combining binning and confidence certificates.

under 3000 steps and runs within minutes. We provide more details on implementation, run times, and convergences in Appendix G.1.

## 4 Adversarial Calibration Training

After introducing the notion of certified calibration, which extends the notion of certification to model calibration, a natural question is whether adversarial training techniques can improve certified calibration. First, we introduce preliminaries and then a novel training method, *adversarial calibration training*, that can improve certified calibration.

**Adversarial Training.** Adversarial training (AT) is commonly used to improve the adversarial robustness of neural networks (Kurakin et al., 2017; Madry et al., 2018). While standard training minimizes the empirical risk of the model under the data distribution, AT minimizes the risk under a data distribution perturbed by adversaries. Salman et al. (2019) propose such an AT method specifically for randomized smoothing models: The SmoothAdv method minimizes the cross-entropy loss, $\mathcal{L}_{CE}$, of a smooth soft classifier $\bar{f}_\theta$ under adversary $\gamma^*$.[3] We note that the SmoothAdv objective can be seen as a constrained optimization problem, minimizing the model risk over some perturbation, conditioning upon that perturbation being a $\mathcal{L}_{CE}$-adversary on $\bar{f}_\theta$. In this work, we have shown the impact of adversaries on calibration and thus propose to minimize the model risk of $\bar{f}_\theta$ under *calibration adversaries*, giving rise to *adversarial calibration training* (ACT).

**Adversarial Calibration Training.** We propose to train models by minimizing the negative log-likelihood under *calibration adversaries*. Let $\mathcal{L}_{\text{Calibration}}$ be a loss on the calibration, and let $\theta \in \Theta$ be the model parameters. We propose the following objective:

$$\min_{\theta \in \Theta} \mathcal{L}_{CE}\left(\bar{f}_\theta\left(\mathbf{x}_n + \gamma_n^*\right), y_n\right) \quad \text{s.t.} \quad \gamma_n^* = \arg\max_{\|\gamma_n\|_2 \leq \epsilon} \mathcal{L}_{\text{Calibration}}\left(\left[\bar{f}_\theta\left(\mathbf{x}_i + \gamma_i\right)\right]_{i=1}^N, \mathbf{y}\right) \quad (7)$$

for *adversarial calibration training* (ACT) in two flavors: Brier-ACT, which uses the Brier score to find adversaries ($\mathcal{L}_{Brier}$-adversaries), and ACCE-ACT, which uses the ACCE to obtain $\mathcal{L}_{ACCE}$-adversaries. While the former is a straightforward extension of SmoothAdv, the second requires more thought, as we describe now.

The $\mathcal{L}_{ACCE}$-adversary can be found by extending the mixed-integer program in (6). The purpose of the mixed-integer program in (6) is to *evaluate* the ACCE, thus maximizing the calibration error over confidences $\mathbf{z}$ and binning $\mathbf{a}$. Here, we seek the adversarial input to the model, $\mathbf{x} + \gamma^*$ to perform ACT and thus reformulate the mixed-integer program w.r.t. $(\mathbf{a}, \gamma)$ instead.

More precisely, we introduce a linear function $h : \mathbb{R}^{N \times K} \to \mathbb{R}^{NM}$ that maps $\{\bar{f}(\mathbf{x}_n + \gamma_n)\}_{n=1}^N \mapsto \mathbf{z}$, where $K$ is the number of classes, $N$ the number of samples, and $M$ the number of bins. This function selects the top confidence from $\bar{f}_\theta$ per data point $n$, then replicates that value $M$ times across all bins and concatenates over $N$ data points (the mapping is $\mathbb{R}^K \to \mathbb{R}^M$ per data point). We formally state the mixed-integer program over $\left(\mathbf{a}, \{\gamma_n\}_{n=1}^N\right)$ with objective

$$\gamma_n^*, \mathbf{a}^* = \arg\max_{\gamma, \mathbf{a}} \|\mathbf{E}\left(h\left(\left[\bar{f}_\theta\left(\mathbf{x}_i + \gamma_i\right)\right]_{i=1}^N\right)\right)^\top \mathbf{a}\|_1 \quad (8)$$

$$\text{subject to:} \quad \mathbf{a} \in \{0,1\}^{NM}, \quad \mathbf{C}^\top \mathbf{a} = \mathbf{1}_N, \quad \mathbf{K}^\top \mathbf{a} = \mathbf{0}_N, \quad \|\gamma_n\|_2 \leq \epsilon.$$

**Implementation Details.** We follow Salman et al. (2019), who solve the objective using SGD and approximate the output of $\bar{f}_\theta$ using Monte Carlo samples. As introduced in Section 3.4, we use ADMM to approximately solve the problem in (8), giving rise to the $\mathcal{L}_{ACCE}$-adversary for ACCE-ACT. We obtain $\mathbf{z}$ through a forward pass and update it as function of updates on $\gamma$. Afterwards, we update other primal variables and perform dual ascent steps. Further, we obtain the constraint variable $\mathbf{K}$ using the Standard **C2** certificate (Kumar et al., 2020) based on very few Monte Carlo samples, as we certify without abstaining. The full algorithm is stated in Appendix F.2.

---

[3]As above, $\bar{f}_\theta$ denotes the model returning $K$ class confidences, whereas $\bar{z}$ returns only the top confidence. As we are updating model parameters, we explicitly add $\theta$ to the model notation in this section.

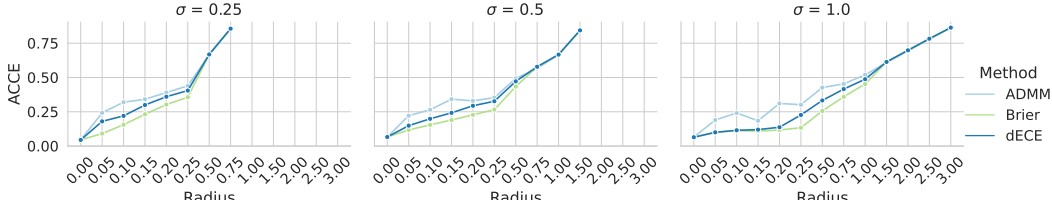

Figure 3: The ACCE returned by ADMM, dECE, and the Brier confidences are shown here for ImageNet. ADMM is the most effective method, as it uniformly yields the largest bounds.

**Complexity.** The complexity of ACCE-ACT in the data dimensionality, $D$, is determined by backprop to update the adversary, $\gamma$, and hence is equal to standard training methods. Further, the objective in (8) is sparse in $\mathbf{C}$, $\mathbf{K}$ and $\mathbf{E}$, enabling the same linear scaling in batch size as the MIP in (6). The computational cost of ACCE attacks is comparable to standard SMOOTHADV (on average 3.4% slower on CIFAR-10 and 2.6% slower on ImageNet). While additional primal and dual updates are required, they are cheap in comparison to the update of the adversary that is obtained by backpropagating through the network.

## 5 EXPERIMENTS

We empirically evaluate the methods introduced above. First, we evaluate the proposed metrics: In Section 5.1 we discuss the certified Brier score, and in Section 5.2 we compare various approaches to approximate the certified calibration error and demonstrate that solving the mixed-integer program using the ADMM solver works best. Second, in Section 5.3, we evaluate adversarial calibration training (ACT) and show it yields improved certified calibration.

### 5.1 CERTIFIED BRIER SCORE

**Experimental Method.** We rely on Gaussian smoothing classifiers to obtain the certifiable radius based on the hard classifier ($\mathbf{C1}$) (Cohen et al., 2019) and bound confidences with the STANDARD certificate ($\mathbf{C2}$). We follow the work on certifying confidences (Kumar et al., 2020) in our experimental setup. We use a ResNet-110 model for CIFAR-10 experiments and a ResNet-50 for ImageNet trained by Cohen et al. (2019). For ImageNet, we sample 500 images from the test set, following prior work. Gaussian smoothing is performed on $100,000$ samples, and we certify at $\alpha = .001$. We only certify calibration at $R$ when the prediction can be certified at $R$. We compute certified metrics on $R \in \{0, 0.05, 0.1, 0.15, 0.2, 0.25, 0.5, 0.75, 1.00, 1.50, 2, 2.5\}$ for CIFAR10 and additionally 3.0 for ImageNet.

**Results.** The certified Brier scores for CIFAR-10 are shown in Figure 16a (Appendix H) and for ImageNet in Figure 2. For both datasets, we observe that the CBSs *increases* with larger certified radii. Models with small $\sigma$ suffer from about a 100% increase in Brier score at $\epsilon = 0.25$, while stronger smoothed models only increase by $< 50\%$. As strongly smoothed models yield tighter certificates on confidences (see Figure 8 in Appendix C), we find that those models are more robust for larger radii at the cost of performance on smaller radii.

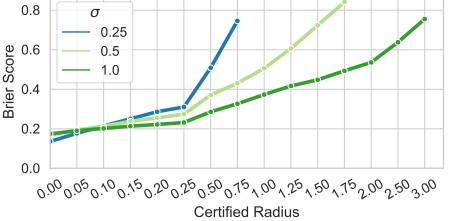

Figure 2: Certified Brier scores on ImageNet. For small radii, small smoothing $\sigma$ outperforms larger ones, but as radii increase, large $\sigma$ outperform smaller $\sigma$.

### 5.2 ACCE

We compare the ACCE solution obtained by ADMM with two baseline and find that ADMM yields better approximations than both. First, we obtain the confidences bounding the Brier score (the "Brier confidences", see (3)) and compute the resulting ECE as baseline. Second, we utilize the *differentiable calibration error* (dECE) (Bohdal et al., 2023) and perform gradient ascent to maximize it (see Appendix M.1). We compare these methods for a range of certified radii and different smoothing $\sigma$.

**Experimental Method.** Our experimental setup is that of Section 5.1 with the difference that we focus on a subset of 2000 certified samples for CIFAR-10.

**Results.** The ACCE returned by ADMM and baselines across certified radii is shown in Figure 15 in Appendix G.3 for CIFAR10 and in Figure 3 for ImageNet. We may observe that large certified radii are associated with worse calibration. We find that ADMM uniformly yields *higher* ACCE than the dECE with differences up to approximately 0.2, which is a strong qualitative difference in calibration. This result suggests that solving the proposed mixed-integer program with ADMM is far more effective than the other methods in approximating the CCE. For larger-scale experiments, see Appendix G.3. Interestingly, we observe that all three methods return ACCE scores that are approximately equal at large radii. We believe this is not an insufficiency of ADMM or the dECE to approximate the CCE, but rather that the CCE and CBS solutions converge. For instance, for accuracies 0 and 1, and unbounded confidences, it is trivial to see that the CCE is achieved by the same confidences as the CBS. We conjecture that this is the case for accuracies in between as well.

## 5.3 Adversarial Calibration Training

We propose to apply ACT as introduced in Section 4 as a fine-tuning method on models that have been pre-trained using SmoothAdv (Salman et al., 2019). We refer to SmoothAdv as standard adversarial training (AT) to disambiguate it from adversarial calibration training (ACT). We demonstrate that our newly proposed ACT methods, Brier-ACT and ACCE-ACT are capable of improving model calibration.

**Experimental Method.** Across all experiments, we compare fine-tuning with Brier-ACT and ACCE-ACT to the AT baseline. For the majority of the experiments, we focus on CIFAR-10 due to the lower computational cost of SmoothAdv training and randomized smoothing. For certified radii $\{0.05, 0.2, 0.5, 1.0\}$, we select 1 to 3 models from Salman et al. (2019) as baselines that provide a good balance between certified calibration and certified accuracy (for baseline metrics and details on the selection procedure, see Appendix I.2). For each baseline,

Table 2: Comparison of calibration for CIFAR-10 models achieving within 3% of the highest certified accuracy (CA) per training method across certified radii (from 0.05 to 1.00). Metrics are the approximate certified calibration error (ACCE ↓) and certified Brier score (CBS ↓).

| Metric | Method | 0.05 | 0.20 | 0.50 | 1.00 |
|--------|--------|------|------|------|------|
| CA | | [83, 86] | [74, 78] | [56, 59] | [35, 38] |
| ACCE | AT | 10.90 | 27.83 | 49.30 | 56.36 |
| ACCE | Brier-ACT | **9.97** | **27.07** | **41.32** | 52.13 |
| ACCE | ACCE-ACT | 10.06 | 27.22 | 42.16 | **47.08** |
| CBS | AT | 8.70 | 10.94 | 28.88 | 36.25 |
| CBS | Brier-ACT | **8.10** | **10.20** | 19.38 | 32.54 |
| CBS | ACCE-ACT | 8.82 | 10.73 | 20.94 | **24.87** |

we perform Brier-ACT and ACCE-ACT for 10 epochs using a set of 16 diverse hyperparameter combinations (more details in Appendix I.3). Regarding data, model architecture, and Gaussian smoothing, we follow the same setup as in Section 5.2. In contrast to Section 5.2, we utilize the CDF certificate on confidences (Kumar et al., 2020) instead of the Standard certificate as these are significantly tighter and thus will be used in practice (see Appendix C a comparison and A.1 for a formal definition).

For each fine-tuned model, we compare the certified accuracy (CA), ACCE, and CBS. During evaluation, the reported ACCE of any model is the maximum out of 16 runs with a diverse grid of hyperparameters (see Appendix I.1). We obtain the CBS in closed form using (3).

**Results.** We find that both flavors of ACT are capable of significantly reducing the certified calibration error while not harming or even improving certified accuracy. In Table 2, we

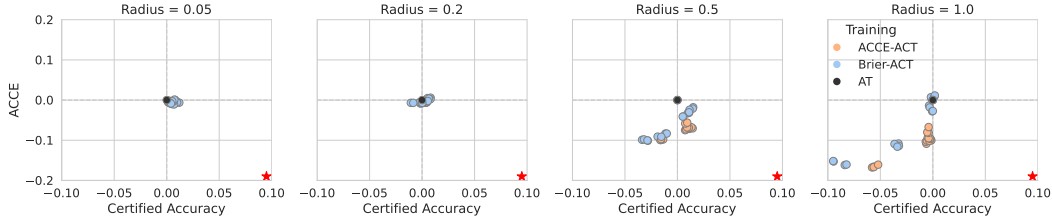

Figure 4: This figure shows the impact of fine-tuning a model via adversarial calibration training (ACT) on *certified accuracy* and *approximate certified calibration error* (ACCE). Each sub-figure presents an adversarial training baseline ("AT") at the origin, certified at the given radius. Following multiple fine-tunings using Brier-ACT and ACCE-ACT, changes in metrics are depicted. The ideal corner is indicated by ⋆. ACCE-ACT significantly improves certified calibration at larger radii.

compare all models that come within 3% of the highest certified accuracy (CA) achieved by any model at a given radius, and evaluate the best certified calibration of these models. While the effect of ACT is very small on small radii ($R = 0.05$ to $R = 0.2$), for larger radii, we observe an improved ACCE (down to $-9.3\%$) and CBS (down to $-11.4\%$), both of which are qualitatively strong differences. In particular, we find that certified calibration of ACCE-ACT at a radius of 1.0 is better than the certified calibration of AT at a radius of 0.5. Comparing both flavors of ACT, we find that ACCE-ACT outperforms Brier-ACT on $R = 1.0$ while performing similarly on smaller radii. Further, the effect of ACCE-ACT on the CBS is strong, which indicates that the calibration of models trained with ACCE-ACT generalizes across calibration metrics. We find that the rank correlation between the ACCE and the CBS across all models is $\rho = 0.98$, indicating a strong agreement between metrics.

In this work, we have repeatedly argued for the importance of considering calibration besides accuracy and as a result, selecting the most appropriate model is now a multi-objective decision problem. To further illustrate this, we pick the best baseline model per certified radius and plot the changes in ACCE against certified accuracy for each fine-tuned model in Figure 4. In these figures, the top left quadrant is strictly dominated by the pre-trained model, while models in the lower right corner improve upon accuracy and calibration (indicated by $\star$). Using ACT, we are able to *jointly* improve upon certified accuracy and calibration. In line with the results in Table 2, improvements on small radii are limited. However, for a certified radius of 0.5, we observe decreases in ACCE of 7.5% while increasing the accuracy by 1.5 %. For a radius of 1.0, we observe an improvement of the ACCE of 12% without harming accuracy. This is a strong qualitative difference. In Appendix I.4, we replicate Figure 4 for more baseline models and demonstrate that improvements are also possible on baselines with better certified calibration. Further, Appendix I.5 provides ablation studies showing that fine-tuning with AT cannot achieve the same calibration as ACT.

We replicate these experiments on FashionMNIST, SVHN, CIFAR-100, and ImageNet, and present results in Appendix J. We demonstrate consistent improvements across datasets with the exception of ImageNet, for which only marginal effects are visible (1-2%). This is in line with prior work optimizing calibration, which does not report ImageNet (Bohdal et al., 2023; Obadinma et al., 2024) or struggles to find consistent effects (Karandikar et al., 2021). See Appendix K for a discussion of trade-offs between calibration and robustness.

## 6 RELATED WORK

Only few papers discuss the confidence scores on certified models. Jeong et al. (2021) propose a variant of *mixup* (Zhang et al., 2018a) for certified models to reduce overconfidence in runner-up classes with the goal of increasing the certified radius, but do not examine confidence as uncertainty estimator. Others propose using conformal predictions on certified models to obtain certifiable prediction sets (Hechtlinger et al., 2019; Gendler et al., 2022). A wider body of literature relates adversarial robustness to calibration on non-certified models. Grabinski et al. (2022) show that robust models are better calibrated, while other work shows that poorly calibrated data points are easier to attack (Qin et al., 2021). This is used by the latter to improve calibration through adversarial training. Stutz et al. (2020), utilize confidence scores and calibration techniques to improve adversarial robustness. Few works investigate the calibration of uncertainty calibration under adversarial attack (Sensoy et al., 2018; Tomani and Buettner, 2021; Kopetzki et al., 2021), however, their work is not necessarily applicable here, as these do not study certified models.

While some works provide bounds on calibration in various contexts, none of them are applicable to our setup (Kumar et al., 2019; Qiao and Valiant, 2021; Wenger et al., 2020). Further, few works have proposed to directly optimize calibration. Kumar et al. (2018) propose a kernel-based measure of calibration that they use to augment cross-entropy training to effectively reduce calibration error. Bohdal et al. (2023) develop the dECE and use meta-learning to pick hyperparameters during training to reduce the dECE on a validation set. A similar metric been proposed by Karandikar et al. (2021) as SB-ECE alongside S-AvUC that increases uncertainty on incorrect samples and decreases uncertainty on correct samples. Similarly, Obadinma et al. (2024) use a combination of surrogate losses to attack calibration. However, they do not directly train on calibration metrics.

ACKNOWLEDGMENTS

This work is supported by a UKRI grant Turing AI Fellowship (EP/W002981/1). C. Emde is supported by the EPSRC Centre for Doctoral Training in Health Data Science (EP/S02428X/1) and Cancer Research UK (CRUK). A. Bibi has received an Amazon Research Award. F. Pinto's PhD is funded by the European Space Agency (ESA). T. Lukasiewicz is supported by the AXA Research Fund. The authors also thank the Royal Academy of Engineering and FiveAI. In addition, the authors thank Mohsen Pourpouneh for his support.

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

# A  PREREQUISITES

## A.1  GAUSSIAN SMOOTHING

As some readers might not be familiar with certification and Gaussian smoothing, we provide a brief introduction here. However, before getting into the details of Gaussian smoothing, we restate some notation introduced above and reiterate the certification assumptions from Section 2.2 for readability.

**Certification.** Let $f : \mathcal{X} \to \Delta_K$ be any model outputting a $K$-dimensional probability simplex over labels $\mathcal{Y}$. We denote the $k$-th component of $f$ as $f_k$. We obtain the classifier indicating the predicted label $F : \mathcal{X} \to \mathcal{Y}$ as the maximum of the softmax: $F(\mathbf{x}) = \arg\max_{k \in \mathcal{Y}} f_k(\mathbf{x})$. We refer to this as the *hard classifier*. We denote the function indicating the *confidence* of this prediction using $z : \mathcal{X} \to [0, 1]$, which we obtain by $z(\mathbf{x}) = \max_{k \in \mathcal{Y}} f_k(\mathbf{x})$. We refer to this as the *soft classifier*.

Our method assumes two certificates.

**Assumption A.1.** Model predictions are invariant under all perturbations,

$$\forall \|\boldsymbol{\gamma}_n\| \le R_n : \quad F(\mathbf{x}_n + \boldsymbol{\gamma}_n) = F(\mathbf{x}_n). \tag{C1}$$

**Assumption A.2.** Prediction confidences are bounded under perturbations,

$$\forall \|\boldsymbol{\gamma}_n\| \le R_n : \quad l\left(z(\mathbf{x}_n), R_n\right) \le z(\mathbf{x}_n + \boldsymbol{\gamma}_n) \le u(z(\mathbf{x}_n), R_n). \tag{C2}$$

We further, assume that $R_n$, $l(\cdot)$ and $u(\cdot)$ can be computed for each sample $\mathbf{x}_n$.

**Gaussian Smoothing.** We achieve these guarantees using Gaussian smoothing (Cohen et al., 2019). Gaussian smoothing scales even to large datasets such as ImageNet and is agnostic towards model architecture. First, we obtain certificate **C1**, explain some nuances of Gaussian smoothing and then introduce certificate **C2**.

Gaussian smoothing takes any arbitrary base model $F$ and constructs a certified model (Cohen et al., 2019). The *smooth hard classifier*, $\bar{F}$, is obtained by:

$$\bar{F}(\mathbf{x}_n) = \arg\max_{c \in \mathcal{Y}} \mathbb{P}_{\boldsymbol{\delta}}(F(\mathbf{x}_n + \boldsymbol{\delta}_n) = c). \tag{9}$$

where $\boldsymbol{\delta}_n$ is sampled from an isotropic Gaussian, i.e. $\boldsymbol{\delta}_n \sim N(0, \sigma^2 \mathbf{I}_D)$. Cohen et al. (2019) show the certificate **C1** for $\bar{F}$ with a radius $R_n$ given by

$$R_n = \frac{\sigma}{2} \left( \Phi^{-1}(p_n^A) - \Phi^{-1}(p_n^B) \right), \tag{10}$$

where $\Phi$ denotes the Gaussian CDF, $p_n^A$ denotes the probability of predicting class $A$, i.e., $\mathbb{P}_{\boldsymbol{\delta}}(F(\mathbf{x}_n + \boldsymbol{\delta}_n) = c_A)$, and $p_n^B$ denotes the probability of predicting the runner-up class: $\max_{B \ne A} \mathbb{P}_{\boldsymbol{\delta}}(F(\mathbf{x}_n + \boldsymbol{\delta}_n) = c_B)$.

The true $\bar{F}$, $p_n^A$ and $p_n^B$ are unknown and have to be estimated. The authors propose to use a finite number of Gaussian samples to estimate $\bar{F}(\mathbf{x}_n)$, a lower bound $\underline{p_n^A}$ on $p_n^A$ and a an upper bound $\overline{p_n^B}$ on $p_n^B$, which are used to estimate certifiable radius $\hat{R}_n$. We choose a significance level $\alpha \in (0, 1)$ in estimating the model (usually $\alpha = .001$). With probability larger $1 - \alpha$, the true model $\bar{F}$ certifiably predicts the same class as the estimated model with a true radius of $R_n \ge \hat{R}_n$. If the evidence is insufficient to certify a prediction at threshold $\alpha$, the model abstains.

This work has been extended by Kumar et al. (2020) to certify confidences, i.e. obtaining certificate **C2**. We denote the *smooth model* as

$$\bar{f}(\mathbf{x}_n) = \mathbb{E}_{\boldsymbol{\delta}}[f(\mathbf{x}_n + \boldsymbol{\delta}_n)], \tag{11}$$

from which we obtain the *smooth soft classifier* as $\bar{z}(\mathbf{x}_n) = \max_{k \in \mathcal{Y}} \bar{f}_k(\mathbf{x}_n)$. Kumar et al. (2020) propose two certificates that we apply on $\bar{z}$, which we refer to as the STANDARD

certificate and the CDF certificate as mentioned above. The STANDARD certificate is very closely related to the certificate given by Cohen et al. (2019) and bounds the confidences as

$$\Phi_\sigma(\Phi_\sigma^{-1}(\underline{\bar{z}(\mathbf{x}_n)}) - R) \leq \bar{z}(\mathbf{x}_n + \boldsymbol{\gamma}_n) \leq \Phi_\sigma(\Phi_\sigma^{-1}(\overline{\bar{z}(\mathbf{x}_n)}) + R) \tag{12}$$

given a radius $R$, where $\underline{\bar{z}(\mathbf{x}_n)}$ and $\overline{\bar{z}(\mathbf{x}_n)}$ are lower and upper bound on $\bar{z}(\mathbf{x}_n)$ obtained using Hoeffding's inequality, and $\Phi_\sigma$ is the Gaussian CDF with standard deviation $\sigma$. They further propose the CDF certificate. It utilises extra information to tighten the certificate: the eCDF is estimated and the Dvoretzky-Kiefer-Wolfowitz inequality is used to find certificate bounds at threshold $1 - \alpha$. Let $s_0 = 0$ and $s_{n+1} = 1$ and let $s_0 < s_i \leq \ldots \leq s_n < s_{n+1}$ partition the interval $[0,1]$. The bounds are given by

$$\sum_{j=1}^{n} (s_j - s_{j-1}) \, \Phi_\sigma \left( \Phi_\sigma^{-1} \left( \underline{p_{s_j}}(\mathbf{x}_n) \right) - R \right) \leq \bar{z}(\mathbf{x}_n + \boldsymbol{\gamma}_n)$$

$$\leq \sum_{j=1}^{n} (s_{j+1} - s_j) \, \Phi_\sigma \left( \Phi_\sigma^{-1} \left( \overline{p_{s_j}}(\mathbf{x}_n) \right) + R \right), \quad (13)$$

where $\underline{p_{s_j}}$ is the lower bound on $\bar{z}$ in the $j$-th bin of the eCDF, and $\overline{p_{s_j}}$ is the upper bound.

## A.2 OTHER CERTIFICATION METHODS

This paper employs Gaussian smoothing (Appendix A.1) due to its scalability to large models and datasets, competitive certifiable sets, and its prominence in the literature. Nonetheless, a variety of alternative methods exist for obtaining certificates as defined in Equations **C1** and **C2**. Certification in machine learning is closely related to the broader field of *formal verification*, which focuses on proving system correctness.

One significant branch is abstract interpretation, which includes techniques such as interval-bound propagation (IBP) (Mirman et al., 2018; Gowal et al., 2019), CROWN (Zhang et al., 2018b), its variant $\beta$-CROWN (Wang et al., 2021), and DeepPoly (Singh et al., 2019). Other methods in this category leverage satisfiability modulo theories (SMT) solvers (Ehlers, 2017; Katz et al., 2019; 2022).

Relaxation-based techniques, such as Neurify (Wang et al., 2018), approximate non-linear constraints with computationally efficient relaxations to enable certification. Certified training has also emerged as a crucial direction to improve model robustness by integrating certification constraints into the training process, resulting in models that are inherently easier to certify. For instance, Gowal et al. (2019) enhance IBP-based certified training by improving bound tightness during training, building on the foundational work of DiffAI (Mirman et al., 2018), which introduced differentiable abstract interpretation for robustness.

Methods such as COLT (Balunovic and Vechev, 2020) and SABR (Müller et al., 2023) focus on improving adversarial robustness while achieving strong certification performance. While these methods do not exclusively target certified training, they represent significant progress in combining adversarial robustness and certification.

This overview, while not exhaustive, highlights key approaches in the field. For comprehensive reviews, readers are referred to Liu et al. (2021), which covers verification algorithms, Silva and Najafirad (2020), which highlights practical challenges in robustness, and Meng et al. (2022), which provides insights into adversarial training and certification.

## A.3 MIXED INTEGER PROGRAMS

For readers unfamiliar with mixed-integer programming we state the canonical form of a mixed-integer program here.

A mixed-integer program (MIP) is a constrained optimization problem over variables ($x \in \mathbb{R}^N, y \in \mathbb{Z}^M$) that can be written in the following form:

$$\min_{x,y} f(x,y) \tag{14}$$

subject to:

$$g_i(x, y) \leq 0 \quad \text{for } i = 1, 2, \ldots, K$$
$$h_j(x, y) = 0 \quad \text{for } j = 1, 2, \ldots, L$$

In our case the constraints are linear, leading to constraint statements such as

$$Ax = b, \quad Cy = d, \quad \ldots, \tag{15}$$

which are introduced in Section 3.3 after the statement of Theorem 3.5.

## B MOTIVATION

### B.1 CALIBRATION $\neq$ ACCURACY

It is important to note that accuracy and calibration measure different concepts, and one may not infer model performance from calibration with certainty or vice versa. Consider the following examples, where we fix one quantity and construct datasets resulting in other quantity taking on opposing values. These are illustrated in Figure 5. First, we fix the calibration error to $ECE = 0.5$ and for *Case 1*, we construct $N$ data points with label $y_n = 1$, confidence $z_n = 0.5$ and thus prediction $\hat{y}_n = 1$. The calibration error here is 0.5 (represented by the horizontal line between $(z = 0.5, y = 1)$ and $(z = 1, y = 1)$) and the accuracy is 1. For *Case 2*, our predictions remain the same, but we change the labels to $y_n = 0$ resulting in an accuracy of 0 while keeping the calibration error of 0.5. Next, we fix the accuracy to 1 and construct examples with $ECE = 0.5$ and $ECE = 0$. The former is given by *Case 1*. The latter (*Case 3*) can be constructed using $y_n = 1$ and $z_n = 1$. Thus, we can construct a distribution over $Z$ and $Y$ such knowing the accuracy tells us nothing about the calibration error and vice versa. Clearly, when evaluating the quality of predictions, it is insufficient to only assess the accuracy. Hence, we argue that certifying accuracy in safety relevant applications is insufficient and calibration should be considered.

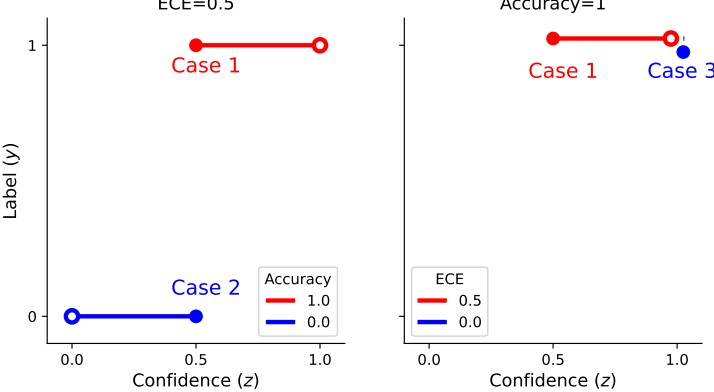

Figure 5: This visualization shows that we can fix either the accuracy or the calibration and construct a dataset to obtain the other quantity with opposite values. The example looks at a binary classification problem. The empty circle displays the point of perfect calibration and the full circle is the calibration on the data. The distance of the line in between is the calibration error.

### B.2 EMPIRICAL ATTACKS

**Training.** We use publicly available checkpoints for the "standard training" model on ImageNet. In particular, we use the PyTorch pretrained v1 model weights. For CIFAR-10 experiments we train a PreActResNet18 model from scratch for 100 epochs. For the robust models we utilize standard adversarial training using PGD and train from scratch for CIFAR-10 while fine-tuning for ImageNet. We use an increase of adversarial budget over 25 epochs to 3/255 and train for 100 epochs for CIFAR-10. For ImageNet we use 10 epochs to warm-up the adversarial budget and train for 50 epochs.

**Attack.** We perform attacks using the criterion described in (2). Using PGD we enforce the constraint on $\|\gamma\|_\infty \leq \epsilon$. Before applying a gradient update,, we check if the condition $F(x + \gamma) = F(x)$ is still satisfied. If this is violated, we stop the optimization. We use a learning rate of 10 for 100 steps with automatic reduction when the loss plateaus. For further details please refer to the published code.

There are differences to Galil and El-Yaniv (2021) in the theoretical objective and the practical setup. Their ACE attack directly optimises confidence scores to increase them when the model incorrect and decrease them when the model is correct. This is what our $(1, y)$-ACE attack does when setting objective $\kappa$ (in their notation) to be the softmax output. However, the loss is different. While Galil and El-Yaniv directly optimise the confidence score (i.e. the top softmax output), we optimise the entire softmax output using cross-entropy. Despite these smaller differences, in essense, their attacks are a special case of ours. Regarding the implementation there are various smaller differences e.g. in the learning rate scheduling and attack methods. For example, while Galil and El-Yaniv use FGSM, while we use PGD. They manually reduce learning rates when the prediction invariance condition would no longer be satisfied by an update, whereas we use automatic plateauing scheduling. For further details plase refer to Galil and El-Yaniv (2021).

**Reliability diagrams.**

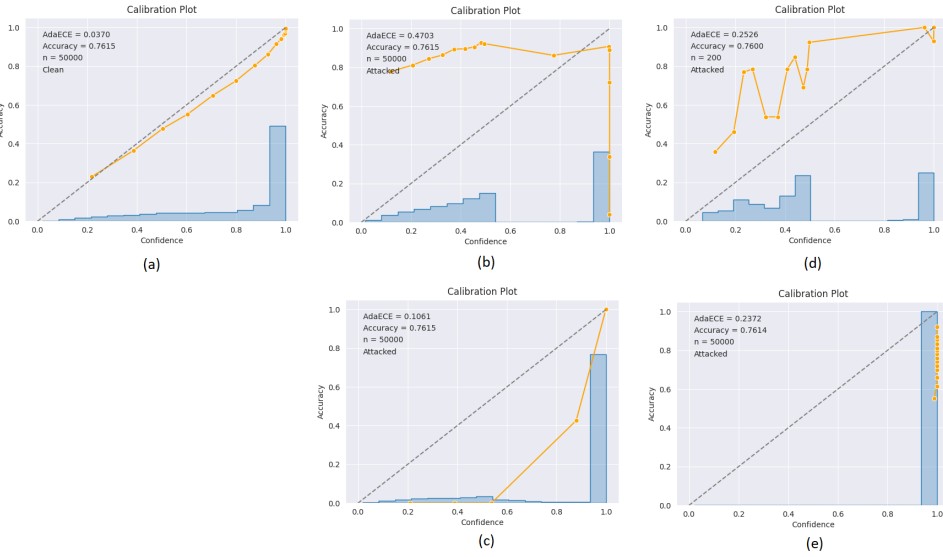

Figure 6: Reliability diagrams for a ResNet50 trained with expected risk minimization on ImageNet-1K and attack radius $\epsilon = 2/255$. (a) No attack, (b) $(1, y)$-ACE attack, (c) $(-1, y)$-ACE attack, (d) $(1, \hat{y})$-ACE attack, (e) $(-1, \hat{y})$-ACE attack. The histogram on the bottom represents the distribution of confidence scores.

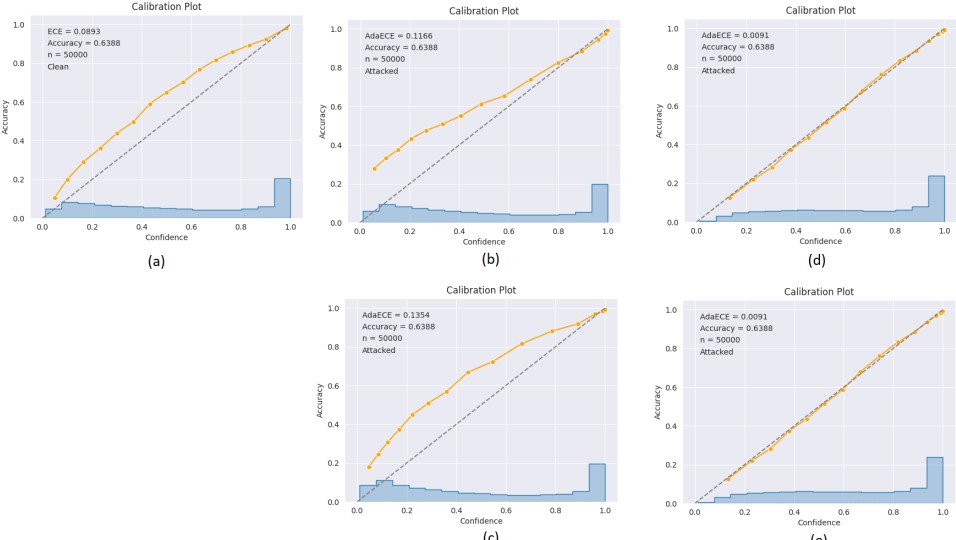

Figure 7: Reliability diagrams for a ResNet50 trained with adversarial risk minimization on ImageNet-1K and attack radius $\epsilon = 2/255$. (a) No attack, (b) $(1, y)$-ACE attack, (c) $(-1, y)$-ACE attack, (d) $(1, \hat{y})$-ACE attack, (e) $(-1, \hat{y})$-ACE attack. The histogram on the bottom represents the distribution of confidence scores.

## C  Confidence Bounds

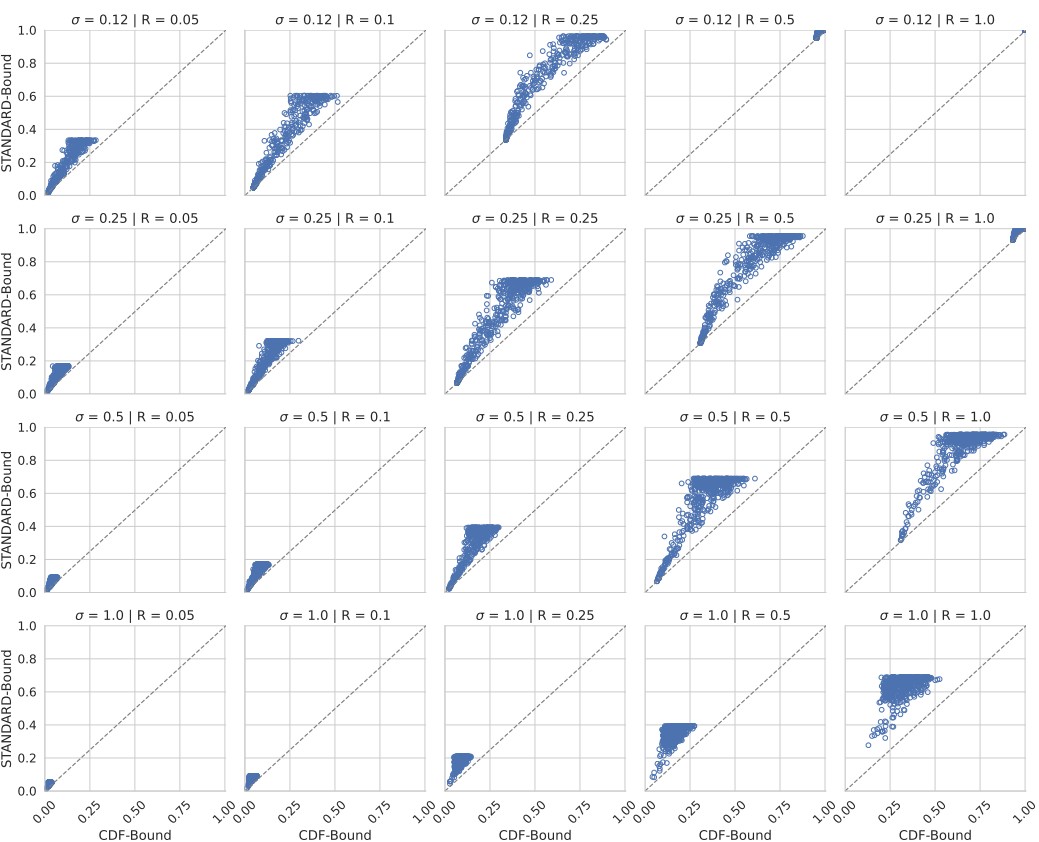

Figure 8: Distance between upper and lower bound of the confidence certificates provided by Kumar et al. (2020). Sub-sampled to 500 samples from the test set of CIFAR10.

.

Kumar et al. (2020) introduce bounds on the confidence score in order to issue a certificate on the prediction, given the lower bound. Therefore, they do not investigate the upper bounds on the confidence. Here, we compute the interval of certified confidences and compare the two certificates on the confidence: The standard STANDARD bound as given in (12) and the more advanced CDF bound. In Figure 8, the distance between the upper and lower bound is plotted for a range of smoothing $\sigma$ and certification radii $R$ on a random subset of 500 samples from the CIFAR-10 test set. The underlying models are trained by Cohen et al. (2019). We may observe that the CDF method yields uniformly tighter bounds in practice.

## D  Brier Bound

Here we provide a proof to Theorem 3.3 as stated above.

*Proof.* Let $\mathbb{I}\{\cdot\}$ be an element-wise indicator function. Assume $\mathbf{z}^* = \mathbf{l}\mathbb{I}\{\mathbf{c} = 1\} + \mathbf{u}\mathbb{I}\{\mathbf{c} = 0\}$ is not the maximum and we want to change $\mathbf{z}^*$ to maximise the TLBS. For some data point with $c_n = 1$, the proposed confidence is the bound $z_n^* = l_n$. Reducing $z_n^*$ is not possible without leaving its feasible set ($l_n \leq z_n^* \leq u_n$), and thus, the only way to find the maximum is to increase it. However, increasing $z_n^*$ would reduce $c_n - z_n^*$, which in turn reduces the error, as $\| \cdot \|_2$ is strictly increasing in $c_n - z_n$. Thus, we have a contradiction for $c_n = 1$. The other case, $c_n = 0$ is analog and both are true for all $n$ and thus, the maximum is proven. $\square$

# E   Calibration as Mixed-Integer Program

## E.1   Restating the Calibration Error

Here, we provide a proof of Theorem 3.5.

*Proof.* We plug (1) into (4) and show equality to the maximum over (5). Let $\mathbf{E}(\mathbf{z})$ and $\mathbf{a}$ be as defined as above and denote $d_n = c_n - z_n$ the gap between confidence and correctness. Further, let $\mathbf{e}_m$ denote the $m$-th column of $\mathbf{E}$.

$$\max_{\forall n: \|\boldsymbol{\gamma}_n\|_2 \leq R} \widehat{\mathrm{NECE}} \left( [\bar{z}(\mathbf{x}_n + \boldsymbol{\gamma}_n)]_{n=1}^N, \mathbf{y} \right) = \max_{\mathbf{l} \leq \mathbf{z} \leq \mathbf{u}} \widehat{\mathrm{NECE}} \left( \mathbf{z}, \mathbf{y} \right) \tag{16}$$

$$= \max_{\mathbf{l} \leq \mathbf{z} \leq \mathbf{u}} \sum_{m=1}^M |B_m| \left| \frac{1}{|B_m|} \sum_{n \in B_m} c_n - \frac{1}{|B_m|} \sum_{n \in B_m} z_n \right| \tag{17}$$

$$= \max_{\mathbf{l} \leq \mathbf{z} \leq \mathbf{u}} \sum_{m=1}^M \left| \sum_{n \in B_m} c_n - z_n \right| \tag{18}$$

$$= \max_{\mathbf{l} \leq \mathbf{z} \leq \mathbf{u}} \sum_{m=1}^M \left| \sum_{n \in B_m} d_n \right| \tag{19}$$

$$= \max_{\mathbf{l} \leq \mathbf{z} \leq \mathbf{u}} \sum_{m=1}^M \left| \sum_{n=1}^N e_{n,m} a_{n,m} \right| \tag{20}$$

$$= \max_{\mathbf{l} \leq \mathbf{z} \leq \mathbf{u}} \sum_{m=1}^M \left| \mathbf{e}_m(\mathbf{z})^\top \mathbf{a} \right| \tag{21}$$

$$= \max_{\mathbf{l} \leq \mathbf{z} \leq \mathbf{u}} \| \mathbf{E}^\top \mathbf{a} \|_1 \tag{22}$$

The equality in (16) is given as we translate the constraint on $[\boldsymbol{\gamma}_n]_{n=1}^N$ into the confidence constraint, $\mathbf{l} \leq \mathbf{z} \leq \mathbf{u}$. The equality assumes the certificate is tight. The equality between (19) and (20) is a result of the unique assignment and valid assignment constraints.

Dividing both sides by $N$ yields the result. □

## E.2   Numerical Example for MIP

In this section, we state a simple numerical example that illustrates the derivation of the Mixed Integer Program (MIP) as presented in Section 3.3. We first define the parameters of this example and then show what values each variable defined in Section 3.3 takes on.

**Setup.**   We consider a problem of $N = 2$ samples predicting classes $\mathcal{Y} = \{C_A, C_B\}$.

Table 3: Example data for numerical example.

| $n$ | $\bar{F}(\mathbf{x}_n)$ | $y_n$ | $l(\bar{z}(\mathbf{x}_n), R)$ | $\bar{z}(\mathbf{x}_n)$ | $u(\bar{z}(\mathbf{x}_n), R)$ |
|-----|------|-------|------|------|------|
| 1 | $C_A$ | $C_A$ | 0.1 | 0.23 | 0.6 |
| 2 | $C_A$ | $C_B$ | 0.5 | 0.78 | 0.9 |

We regard the expected calibration error with $M = 3$ bins and equal-width binning. Hence, the bins are $[0, 1/3)$, $[1/3, 2/3)$, and $[2/3, 1]$.

**MIP variables.** We obtain the vector of correctness for each prediction: $\mathbf{c} = [1 \quad 0]^\top$. Next, we obtain the vector $\mathbf{l}$ and $\mathbf{u}$. To this end, we first obtain $\mathbf{l}^z$ and $\mathbf{u}^z$ from the table above: $\mathbf{l}^z = [0.1 \quad 0.5]^\top$ and $\mathbf{u}^z = [0.6 \quad 0.9]^\top$. We further obtain $\mathbf{l}^B$ and $\mathbf{u}^B$ from the binning scheme of the ECE. $\mathbf{l}^B = [0 \quad 1/3 \quad 2/3]^\top$ and $\mathbf{u}^B = [1/3 \quad 2/3 \quad 1]^\top$. Using these we define $\mathbf{l}$ and $\mathbf{u}$:

$$\mathbf{l} = [0.1 \quad 1/3 \quad 2/3 \quad 0.5 \quad 0.5 \quad 2/3]^\top ,$$
$$\mathbf{u} = [1/3 \quad 0.6 \quad 0.6 \quad 1/3 \quad 2/3 \quad 0.9]^\top . \tag{23}$$

It should become clear that bin $m = 3$ is inaccessible to sample $n = 1$ and bin $m = 1$ is inaccessible to sample $n = 2$. Hence, $\mathbf{k} = [0 \quad 0 \quad 1 \quad 1 \quad 0 \quad 0]^\top$. Using this we define $\mathbf{l}'$ and $\mathbf{u}'$:

$$\mathbf{l}' = [0.1 \quad 1/3 \quad 0 \quad 0 \quad 0.5 \quad 2/3]^\top$$
$$\mathbf{u}' = [1/3 \quad 0.6 \quad 0 \quad 0 \quad 2/3 \quad 0.9]^\top \tag{24}$$

.

We define the vector of confidences $\mathbf{z}$ with inaccessible elements set to 0 (see "Confidence Constraint"). The vector of confidences is not uniquely defined through the problem above. One example is $\mathbf{z} = [0.23 \quad 0.45 \quad 0 \quad 0 \quad 0.6 \quad 0.78]^\top$ together with $\mathbf{a} = [1 \quad 0 \quad 0 \quad 0 \quad 0 \quad 1]^\top$.

We now obtain $e_{n,m} = c_n - z_{n,m}$ to define $\mathbf{e} = [0.77 \quad 0.55 \quad 1 \quad 0 \quad 0.6 \quad 0.78]$. Finally, we obtain $\mathbf{E}$ by stacking $\mathbf{e}$ appropriately.

$$\mathbf{E} = \begin{bmatrix} e_{1,1} & 0 & 0 \\ 0 & e_{1,2} & 0 \\ 0 & 0 & e_{1,3} \\ e_{2,1} & 0 & 0 \\ 0 & e_{2,2} & 0 \\ 0 & 0 & e_{2,3} \end{bmatrix} = \begin{bmatrix} 0.77 & 0 & 0 \\ 0 & 0.55 & 0 \\ 0 & 0 & 1 \\ 0 & 0 & 0 \\ 0 & 0.6 & 0 \\ 0 & 0 & 0.78 \end{bmatrix} \tag{25}$$

The last variable that is value dependent is the matrix $\mathbf{K}$, which we can obtain from $\mathbf{k}$ as follows:

$$\mathbf{K} = \begin{bmatrix} k_{1,1} & 0 \\ k_{1,2} & 0 \\ k_{1,3} & 0 \\ 0 & k_{2,1} \\ 0 & k_{2,2} \\ 0 & k_{2,3} \end{bmatrix} = \begin{bmatrix} 0 & 0 \\ 0 & 0 \\ 1 & 0 \\ 0 & 1 \\ 0 & 0 \\ 0 & 0 \end{bmatrix} \tag{26}$$

### E.3 LAGRANGIAN

We follow Wu and Ghanem (2019) and Bibi et al. (2023) to reformulate the binary-constraints on $\mathbf{a}$. Note that, $\mathbf{a} \in \{0,1\}^{NM} \Leftrightarrow \mathbf{a} \in S_b \cap S_2$, where $S_b$ is the unit hypercube and $S_2$ is the $\ell_2$-sphere, both centered at $\frac{1}{2}$. We introduce auxiliary variables $\mathbf{q}_1 \in S_b$ and $\mathbf{q}_2 \in S_2$ and add constraints $\mathbf{a} = \mathbf{q}_1$ and $\mathbf{a} = \mathbf{q}_2$. Similarly, we replace the constraint $\mathbf{z} \in S_z$ by enforcing it on $\mathbf{g}$ and adding $\mathbf{z} = \mathbf{g}$. Updates on the primal variables $(\mathbf{a}, \mathbf{z}, \mathbf{q}_1, \mathbf{q}_2, \mathbf{g})$ are performed via gradient descent.

We formally state the augmented Lagrangian. The variables $\mathbf{a}, \mathbf{z}, \mathbf{q}_1, \mathbf{q}_2$, and $\mathbf{g}$ are described above, each of dimension $NM$.

$$
\begin{aligned}
\mathcal{L}(\mathbf{a}, \mathbf{z}, \mathbf{g}, \mathbf{q}_{1,2}, \boldsymbol{\lambda}_{1,2,3,4,5}) = & -|\mathbf{E}(\mathbf{z})^\top \mathbf{a}| \mathbf{1}_M \\
& + \mathbb{I}_\infty\{\mathbf{q}_1 \in S_b\} + \boldsymbol{\lambda}_1^\top [\mathbf{a} - \mathbf{q}_1] + \frac{\rho_1}{2}\|\mathbf{a} - \mathbf{q}_1\|_2^2 \\
& + \mathbb{I}_\infty\{\mathbf{q}_2 \in S_2\} + \boldsymbol{\lambda}_2^\top [\mathbf{a} - \mathbf{q}_2] + \frac{\rho_2}{2}\|\mathbf{a} - \mathbf{q}_2\|_2^2 \\
& + \boldsymbol{\lambda}_3^\top [\mathbf{C}^\top \mathbf{a} - \mathbf{1}_N] + \frac{\rho_3}{2}\|\mathbf{C}^\top \mathbf{a} - \mathbf{1}_N\|_2^2 \\
& + \boldsymbol{\lambda}_4^\top \mathbf{K}^\top \mathbf{a} + \frac{\rho_4}{2}\|\mathbf{K}^\top \mathbf{a}\|_2^2 \\
& + \mathbb{I}_\infty\{\mathbf{g} \in S_z\} + \boldsymbol{\lambda}_5^\top [\mathbf{z} - \mathbf{g}] + \frac{\rho_5}{2}\|\mathbf{z} - \mathbf{g}\|_2^2
\end{aligned}
\tag{27}
$$

with dual variables $\boldsymbol{\lambda}_{1,2,5} \in \mathbb{R}^{NM}$, $\boldsymbol{\lambda}_{3,4} \in \mathbb{R}^N$. Here, $\mathbb{I}_\infty$ is 0 if the statement is true and $\infty$ otherwise. The values of $\rho_i > 0$ are hyperparameters to be tuned.

### E.4 ADMM Updates

We perform $T$ ADMM steps. At each ADMM step we cycle through the primal variables $\mathbf{a}$ and $\mathbf{z}$ and perform gradient descent. For the variables $\mathbf{q}_1$, $\mathbf{q}_2$ and $\mathbf{g}$, we obtain an analytic solution by equating the gradient to 0, solving the equation and projecting the variables into their feasible set. For $\mathbf{q}_1$, the update $\mathcal{U}_{\mathbf{q}_1}$ is given by

$$
\mathbf{q}_1 \leftarrow \text{clamp}_{[0,1]}\left(\frac{\boldsymbol{\lambda}_1}{\rho_1} + \mathbf{a}\right),
\tag{28}
$$

for $\mathbf{q}_2$ the update $\mathcal{U}_{\mathbf{q}_2}$ is given by

$$
\mathbf{q}_2 \leftarrow \frac{1}{2}\mathbf{1} + \frac{\sqrt{NM}}{2} \frac{\frac{\boldsymbol{\lambda}_2}{\rho_2} + \mathbf{a} - \frac{1}{2}\mathbf{1}}{\|\frac{\boldsymbol{\lambda}_2}{\rho_2} + \mathbf{a} - \frac{1}{2}\mathbf{1}\|_2},
\tag{29}
$$

and finally, the update $\mathcal{U}_{\mathbf{g}}$ is given by

$$
\mathbf{g} \leftarrow \text{clamp}_{[\mathbf{l}', \mathbf{u}']}\left(\frac{\boldsymbol{\lambda}_5}{\rho_5} + \mathbf{z}\right).
\tag{30}
$$

The updates on the dual variables $\boldsymbol{\lambda}_i$ are performed through a single gradient ascent step with step size $\rho_i$.

With these definitions, we can formalise the algorithm for the ADMM updates.

---
**Algorithm 1** The ADMM Updates

---
**input** ADMM parameters $\{\alpha_\mathbf{a}, \alpha_\mathbf{z}, \rho_{1,2,3,4,5}\}$, primal variables $\left\{\mathbf{a}^{(0)}, \mathbf{z}^{(0)}, \mathbf{q}_1^{(0)}, \mathbf{q}_2^{(0)}, \mathbf{g}^{(0)}\right\}$
  and dual variables $\{\boldsymbol{\lambda}_i\}_{i=1}^5$
**output** ACCE
  **for** $t = 1$ **to** $T$ **do**
    $\mathbf{a}^{(t)} \leftarrow \mathbf{a}^{(t-1)} - \alpha_\mathbf{a} \nabla_\mathbf{a}\mathcal{L}\left(\mathbf{a}^{(t-1)}, \mathbf{z}^{(t-1)}, \mathbf{g}^{(t-1)}, \mathbf{q}_{1,2}^{(t-1)}, \boldsymbol{\lambda}_{1,2,3,4,5}^{(t-1)}\right)$
    $\mathbf{z}^{(t)} \leftarrow \mathbf{z}^{(t-1)} - \alpha_\mathbf{z} \nabla_\mathbf{z}\mathcal{L}\left(\mathbf{a}^{(t)}, \mathbf{z}^{(t-1)}, \mathbf{g}^{(t-1)}, \mathbf{q}_{1,2}^{(t-1)}, \boldsymbol{\lambda}_{1,2,3,4,5}^{(t-1)}\right)$
    $\mathbf{g}^{(t)} \leftarrow \mathcal{U}_\mathbf{g}\left(\mathbf{z}^{(t)}, \boldsymbol{\lambda}_5^{(t-1)}\right)$
    $\mathbf{q}_1^{(t)} \leftarrow \mathcal{U}_{\mathbf{q}_1}\left(\mathbf{a}^{(t)}, \boldsymbol{\lambda}_1^{(t-1)}\right)$
    $\mathbf{q}_2^{(t)} \leftarrow \mathcal{U}_{\mathbf{q}_2}\left(\mathbf{a}^{(t)}, \boldsymbol{\lambda}_2^{(t-1)}\right)$
    $\boldsymbol{\lambda}_i^{(t)} \leftarrow \boldsymbol{\lambda}_i^{(t-1)} + \rho_i \nabla_{\boldsymbol{\lambda}_i}\mathcal{L}\left(\mathbf{a}^{(t)}, \mathbf{z}^{(t)}, \mathbf{g}^{(t)}, \mathbf{q}_{1,2}^{(t)}, \boldsymbol{\lambda}_{1,2,3,4,5}^{(t-1)}\right)$ **for** $i = 1, 2, 3, 4, 5$
  **end for**

---

# F  Adversarial Calibration Training

## F.1  Lagrangian

We formally state the augmented Lagrangian. The variables $\mathbf{a}$, $\boldsymbol{\gamma}$, $\mathbf{q}_1$, $\mathbf{q}_2$ and are described above, each of dimension $NM$.

$$
\begin{aligned}
\mathcal{L}(\mathbf{a}, \boldsymbol{\gamma}, \mathbf{q}_{1,2}, \boldsymbol{\lambda}_{1,2,3,4}) = {} & -|\mathbf{E}(\mathbf{c}, \boldsymbol{\gamma})^\top \mathbf{a}|\mathbf{1}_M \\
& + \mathbb{I}_\infty\{\mathbf{q}_1 \in S_b\} + \boldsymbol{\lambda}_1^\top[\mathbf{a} - \mathbf{q}_1] + \frac{\rho_1}{2}\|\mathbf{a} - \mathbf{q}_1\|_2^2 \\
& + \mathbb{I}_\infty\{\mathbf{q}_2 \in S_2\} + \boldsymbol{\lambda}_2^\top[\mathbf{a} - \mathbf{q}_2] + \frac{\rho_2}{2}\|\mathbf{a} - \mathbf{q}_2\|_2^2 \\
& + \boldsymbol{\lambda}_3^\top[\mathbf{C}^\top\mathbf{a} - \mathbf{1}_N] + \frac{\rho_3}{2}\|\mathbf{C}^\top\mathbf{a} - \mathbf{1}_N\|_2^2 \\
& + \boldsymbol{\lambda}_4^\top\mathbf{K}^\top\mathbf{a} + \frac{\rho_4}{2}\|\mathbf{K}^\top\mathbf{a}\|_2^2
\end{aligned}
\tag{31}
$$

with dual variables $\boldsymbol{\lambda}_{1,2} \in \mathbb{R}^{NM}$, $\boldsymbol{\lambda}_{3,4} \in \mathbb{R}^N$. Here, $\mathbb{I}_\infty$ is 0 if the statement is true and $\infty$ otherwise. The values of $\rho_i > 0$ are hyperparameters to be tuned.

Note, that this is still a constrained optimization problem in $\boldsymbol{\gamma}$.

## F.2  ACT Algorithm

The algorithm here follows the general notation introduced above. Additionally, we define $\bar{f}_\theta^V$ to be the Monte Carlo approximation to $\bar{f}_\theta$ with $V$ samples and $\bar{F}_\theta^V$ the approximation to $\bar{F}_\theta$. We further denote updates on the variables using $\mathcal{U}$. Updates are performed as in Appendix E.4, if not stated otherwise. Here, we use $\mathcal{U}_{PGD}$ to describe an update by projected gradient descent. In the algorithm below, the correctness, $\mathbf{c}$, and the inaccessibility matrix, $\mathbf{K}$, are explicitly added, as these require a forward pass to be computed.

---

**Algorithm 2** Adversarial Calibration Training - One Batch

---

**input** Data $\{\mathbf{x}_n, y_n\}_{n=1,\ldots,N}^N$, ADMM parameters $\{\alpha_{\boldsymbol{\gamma}}, \alpha_{\mathbf{z}}, \rho_{1,2,3,4}\}$, perturbation bound $\epsilon$, and $\mathcal{L} = ACCE$
**output** $\theta$

$\quad \boldsymbol{\gamma}_n^{(0)} \leftarrow \mathbf{0}$ {Initialise Adversary}
$\quad \mathbf{a}^{(0)} \leftarrow [1/M, 1/M, \ldots, 1/M]$ {Initialise binning vector}
$\quad \mathbf{q}_1^{(0)} \leftarrow \mathbf{a}^{(0)}$ {Initialise for Box Constraint}
$\quad \mathbf{q}_2^{(0)} \leftarrow \mathbf{a}^{(0)}$ {Initialise for Sphere Constraint}
$\quad \boldsymbol{\lambda}_i \leftarrow \mathbf{0}$ for $i = 1, 2, 3, 4$ {Initialise dual variables}
$\quad$ **for** $t = 1$ **to** $T$ **do**
$\quad\quad \mathbf{z}^{(t-1)} \leftarrow \left[ h\left( \bar{f}_\theta^V \left( \mathbf{x}_n + \boldsymbol{\gamma}_n^{(t-1)} \right) \right) \right]_{n=1,\ldots,N}$
$\quad\quad$ **if** $t = 1$ **then**
$\quad\quad\quad$ $\mathbf{K}$ is created as described in section 3.3. Here, as function of $\mathbf{z}^{(0)}$. {This is required for the ACCE constraint.}
$\quad\quad\quad$ $\mathbf{c} \leftarrow \left[ \mathbb{I}\{y_n = \bar{F}_\theta^V(\mathbf{x}_n + \boldsymbol{\gamma}^{(0)})\} \right]_{n=1,\ldots,N}$ {The correctness required for the ACCE loss.}
$\quad\quad$ **end if**
$\quad\quad \boldsymbol{\gamma}^{(t)} \leftarrow \mathcal{U}_{PGD}\left( \boldsymbol{\gamma}^{(t-1)}, \mathbf{a}^{(t-1)}, \mathbf{c}, \epsilon \right)$
$\quad\quad \mathbf{a}^{(t)} \leftarrow \mathcal{U}_{\mathbf{a}}\left( \mathbf{z}^{(t-1)}, \mathbf{q}_1^{(t-1)}, \mathbf{q}_2^{(t-1)}, \boldsymbol{\lambda}_{1,2,3,4}^{(t-1)}, \mathbf{K}, \mathbf{c} \right)$
$\quad\quad \mathbf{q}_1^{(t)} \leftarrow \mathcal{U}_{\mathbf{q}_1}\left( \mathbf{a}^{(t)}, \boldsymbol{\lambda}_1^{(t-1)} \right)$
$\quad\quad \mathbf{q}_2^{(t)} \leftarrow \mathcal{U}_{\mathbf{q}_2}\left( \mathbf{a}^{(t)}, \boldsymbol{\lambda}_2^{(t-1)} \right)$
$\quad\quad \boldsymbol{\lambda}_i^{(t)} \leftarrow \boldsymbol{\lambda}_i^{(t-1)} + \rho_i \nabla_{\boldsymbol{\lambda}_i} \mathcal{L}\left( \mathbf{a}^{(t)}, \mathbf{z}^{(t-1)}, \mathbf{q}_{1,2}^{(t)}, \boldsymbol{\lambda}_{1,2,3,4}^{(t-1)} \right)$ **for** $i = 1, 2, 3, 4$
$\quad\quad$ Update Parameters: $\rho_{1:4}$
$\quad$ **end for**
$\quad \theta \leftarrow \mathcal{U}_\theta\left( \bar{f}_\theta^V\left( \mathbf{x} + \boldsymbol{\gamma}^{(T)} \right) \right)$ {Update model parameters}

---

# G   EXPERIMENTS - ACCE

## G.1   ADMM CONVERGENCE

In this section we present some insights and figures that closer examine the run time and convergence of ADMM on the mixed-integer program (5). Numbers reported in this section are based on CIFAR-10.

**Run Time.** In our experiments, ADMM always converges in under 3000 steps. Our implementation utilises the `torch.sparse` package in version 2.0 (Paszke et al., 2019) and runs in less than 2 minutes for 7000 certified data points and 15 bins on a Nvidia RTX 3090. At convergence, we observe that all constraints are sufficiently met, and hence ADMM provides a feasible solution (see below). We validated a number of optimization hyperparameters and fix those for later experiments (see Appendix G.2). We recommend running ADMM on two different initialization of $\mathbf{z}$, the observed adversary-free confidences, as well as those achieving the certified Brier score (3) and subsequently picking the larger ACCE. We find that the maximum ACCE is achieved well before ADMM has converged. Therefore, we recommend projecting a copy of $\mathbf{a}$ and $\mathbf{z}$ into their feasible set after each step and calculating the ACCE.

**Convergence Diagnostics.** Figure 9 shows the development of various components of the Lagrangian (see (27) in Appendix E.3) across steps. Figure 10 shows how well the constraints are met. Finally, Figure 11 plots parts of the assignment vector, $\mathbf{a}$, across steps. All plots are based on 2000 CIFAR-10 predictions of a model trained by (Cohen et al., 2019) with $\sigma = 0.25$.

More precisely, Figure 9 plots seven metrics across 3000 steps of running ADMM. In the first row, you may observe the Lagrangian (ignoring the $\mathbb{I}_\infty$ components). In the second row, you may observe the soft objective divided by $N$, which is the *relaxed* ECE. Further, we project the solution to exactly satisfy the constraints, the *projected* ECE. The third and fourth plot

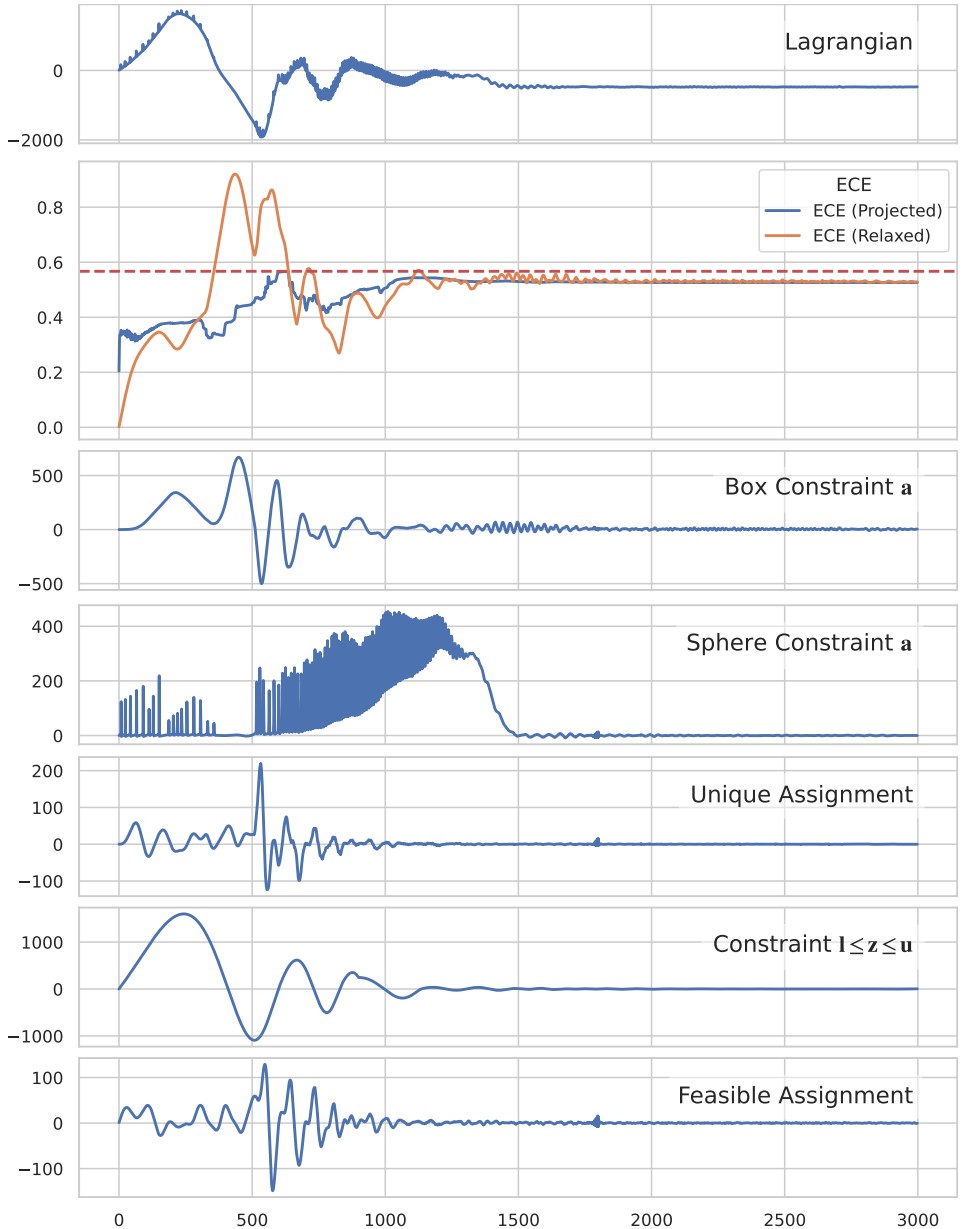

Figure 9: Components of the Lagrangian for ADMM to find the ACCE on CIFAR-10 with $\epsilon = 1.0$. Labels explained in Appendix G.1. Observe, that all constraints are fully met.

are the Box and Sphere Constraints on $\mathbf{a}$ as described above. Together they are equivalent to a binary constraint $\mathbf{a}$. The fifth row plots the constraint on the confidences $\mathbf{l} \leq \mathbf{z} \leq \mathbf{u}$. The sixth row shows the Unique Assignment constraint and the seventh and final row shows the Valid Assignment constraint. You may observe, that all components converge well.

To closer describe Figure 10, we first introduce some notation: For some vector $\mathbf{x}$ and set $S$, we define $d_p(\mathbf{x}, S) := \min_{\mathbf{y} \in S} \|\mathbf{x} - \mathbf{y}\|_p$. Each subplot in Figure 10 tracks one constraint across the ADMM steps. For each constraints we provide $p = 1$ and $p = \infty$ norms for the distance from the constraint being met, i.e. the projection distance to the feasible set. We measure whether the confidence is bounded using $d_p(\mathbf{z}, S_{\mathbf{z}})$ where $S_{\mathbf{z}} = \{\mathbf{z} : \mathbf{l} \leq \mathbf{z} \leq \mathbf{u}\}$. The Binary Assignment constraint is measured by $d_p(\mathbf{a}, \mathcal{A})$, where $\mathcal{A}$ is the set of all binary vectors. Deviation from the Unique Assignment and Valid Assignment constraints are given by $\|\mathbf{C}^\top \mathbf{a} - \mathbf{1}\|_p$, and $\|\mathbf{K}^\top \mathbf{a}\|_p$, respectively.

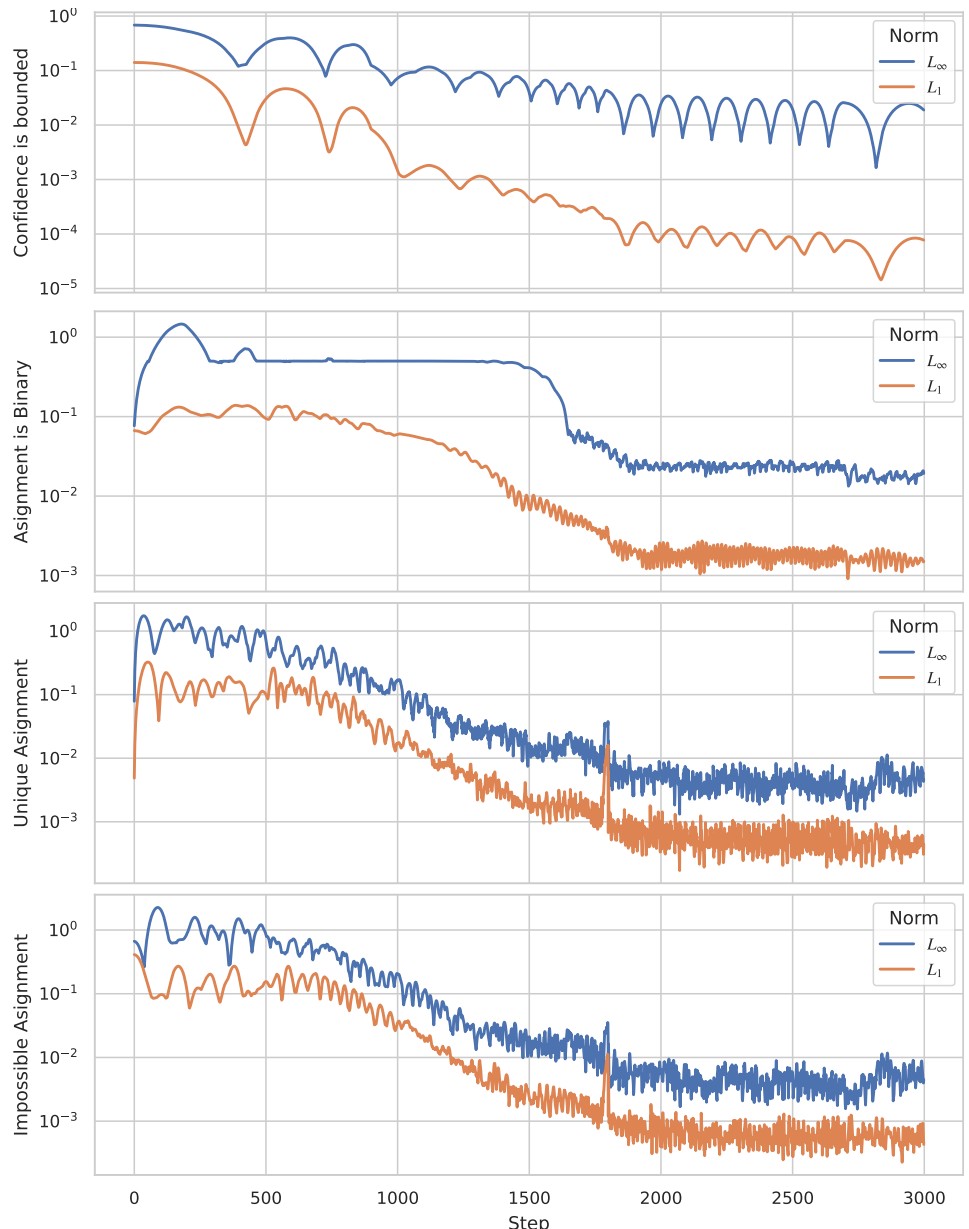

Figure 10: The deviation of constraints of the mixed-integer program given in (6). For precise definitions of the $y$-axis please refer to Appendix G.1.

Finally, we investigate the values of the assignment vector **a** closer. In Figure 11, we plot the first 150 elements of **a** for steps 0, 500 and 3000. These elements are the assignments of 10 data points to 15 bins. As you can see the initialization of **a** is uniform, while a "preference" for some bins becomes apparent at 500 steps and finally, at 3000 steps the vector has converged to a near-perfect binary vector.

## G.2 ADMM HYPERPARAMETER SEARCH

We performed a random hyperparameter search for the ADMM solver to find reasonable hyperparameters that are efficient across experiments.

- The learning rate for **z**, $\alpha_{\mathbf{z}}$: We test values from $1 \times 10^{-5}$ to $1 \times 10^{-2}$ and find it has little influence.

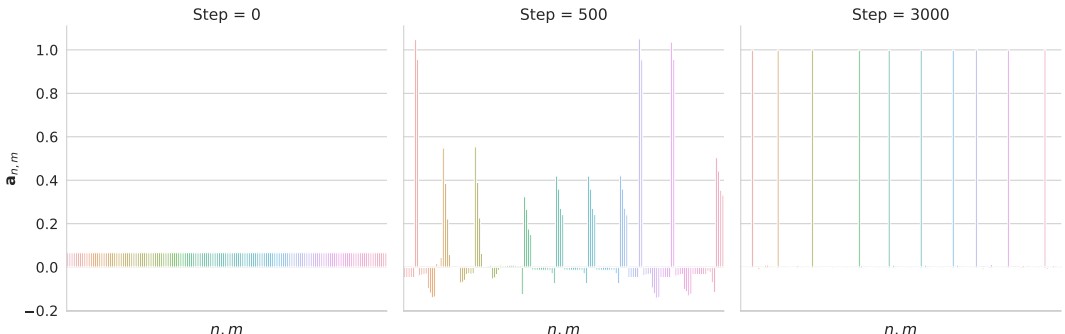

Figure 11: The first 150 elements of the assignment vector in the mixed-integer program across steps of the ADMM solver. The vector is initialized uniform but converges to a binary vector.

- The learning rate for $\mathbf{a}$, $\alpha_{\mathbf{a}}$: We test values from $5 \times 10^{-4}$ to $5 \times 10^{-2}$ for ImageNet and from $1 \times 10^{-5}$ to $1 \times 10^{-2}$ for CIFAR10. Larger learning rates are preferred. In our experiments 0.001 to 0.05 worked best.

- The Lagrangian smoothing variable $\rho_i$: We tested starting values from 0.001 to 0.5. We find little influence but values around 0.01 to 0.05 works best. We test multiplicative schedules to increase $\rho_i$ over time with increases starting at 1‰ to 4%. While larger values of $\rho$ improve constraint convergence, they can dominate gradients when the constraints have not sufficiently converged yet, and thus ADMM easily diverges. Schedules with factors around 1.02 to 1.05 per step are effective and we stop increasing $\rho$ at 10, which is completely sufficient to meet the constraints. We find, that $\rho_i$ scheduling is important. Applying the same schedule described above for all $\rho_1, \ldots, \rho_5$ works reasonably well.

- We test performing 1 to 3 updates for $\mathbf{a}$ and $\mathbf{z}$ per ADMM step and find that more steps do not aid results while slowing down ADMM. We thus, recommend 1 step per ADMM step.

- We test clipping values of $\mathbf{a}$. As we constrain $\mathbf{a} = \mathbf{q}_2$, we clip $\mathbf{a}$ s.t. $\|\mathbf{a} - 1/2\|_\infty \leq 1.2\|\mathbf{q}_2 - 1/2\|_\infty$. Note that when $\mathbf{a}$ converges, it will be significantly smaller than this constraint. We have never observed any decline in performance with this approach but seen that it stabilizes the optimization in rare instances.

- We clip the gradients of $\mathbf{a}$ to 5 in infinity norm aiding stability.

- The initialization of $\mathbf{a}$ has a major effect on the performance of ADMM, more so than any other hyperparameter. While it is an obvious solution to initialize $\mathbf{a}^{(0)}$ such that it is a valid assignment for $\mathbf{z}^{(0)}$, we find that this is majorly outperformed by uniform initializations. We tested 0, $1/M$ and 1 and find that $1/M$ works best.

- The initialization of $\mathbf{z}$ has a mild effect on ADMM performance given that sometimes, we might have prior knowledge on how and in which direction the model currently is miscalibrated. We recommend initializing it with adversary-free confidences and with the confidences achieving the Brier bound and subsequently picking the larger one.

## G.3 ADMM vs. dECE vs. Brier Bound

To further assess the efficacy of the ADMM algorithm, we compare it to an alternative methods of approximating bounds: the *differentiable calibration error* (dECE) (Bohdal et al., 2023) and the bounds obtained by the confidences that maximise the Brier score (*Brier confidences*). We find that ADMM outperforms both other techniques. We performed hyperparameter searches for ADMM and dECE with a wide range, but carefully selected hyperparameters. For ADMM we run between 259 and 1560 trials (depending on the runtime) and for dECE always 2000 trials. We explore a subset of radii and smoothing $\sigma$ (as visible from our plots). Our first observation is, that the dECE is very sensitive to the initialization of confidence scores indicating that gradient ascent on dECE (even with very large learning

rates) does not sufficiently explore the loss surface. While differences are also observable for ADMM, these are very small. Figure 12 show these differences for ImageNet.

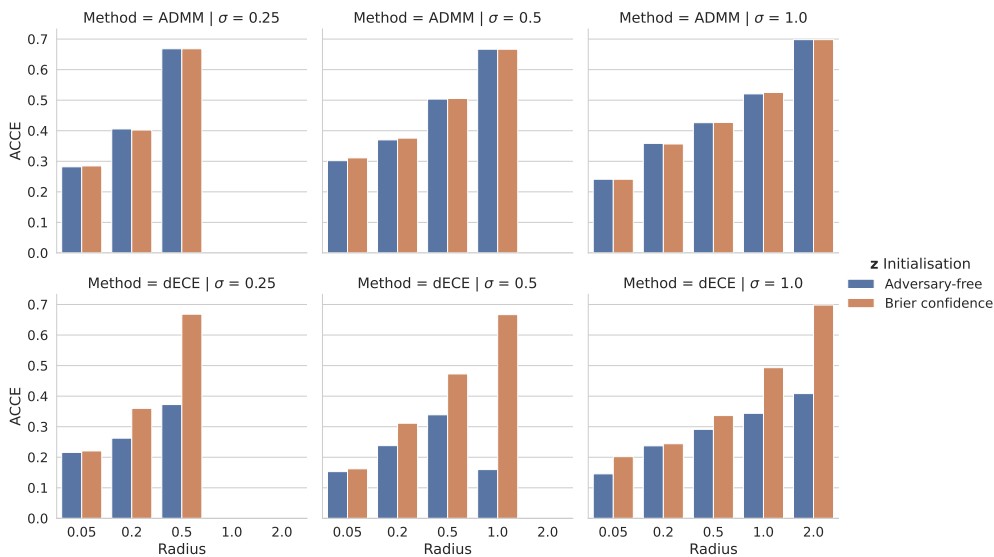

Figure 12: The results of the hyperparameter search are shown here. For each combination of dataset, $\sigma$, radius, and *initialization* of **z**, the maximum achieved by the three methods, ADMM, dECE and Brier are shown here. ADMM performs well regardless of the initialization of **z**. The dECE is depends highly on the initialization.

Second, as a result of our hyperparameter search, we note that the best ADMM results outperform the best dECE by a significant margin. To demonstrate this, we compare the maxima achieved across trials by the dECE, the Brier bound and ADMM in Figure 13.

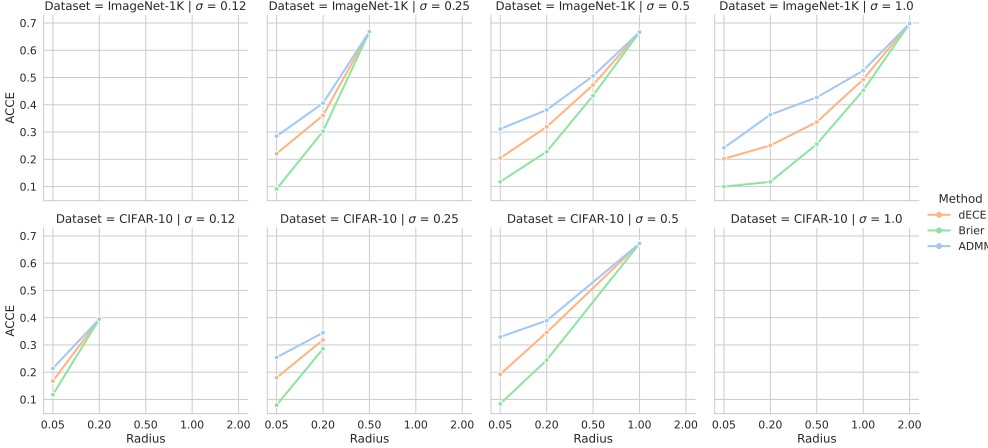

Figure 13: The results of the hyperparameter search are shown here. For each combination of dataset, $\sigma$, and radius, the maximum achieved by the three methods, ADMM, dECE and Brier are shown here. While dECE is able to be uniformly better than the Brier confidences, ADMM outperforms both by a significant margin.

As noted in the main paper, for large radii, the methods converge to each other. Beyond the comparison of maximum values above, we find that even a single *one-size-fits-all* set of hyperparameters for ADMM outperforms the maximum achieved by the dECE hyperparameter search in most cases as shown in Figure 14. The hyperparameters used for ADMM are as follows: All $\rho_i$s are initialised to 0.01 with increases every step by a factor of 0.4%. Both learning rates are set to 0.001. The assignments are initialised to $1/M$, the confidence is

initialised to adversary-free and Brier confidence, a measure benefiting the dECE much more than the ADMM. The gradients of the assignment variable **a** are clipped to 1.

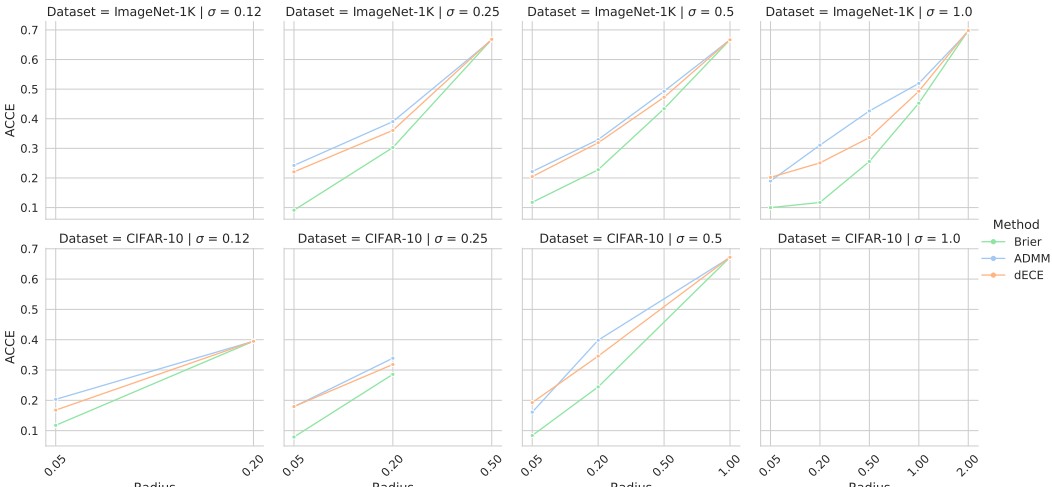

Figure 14: We compare a single set of parameters for ADMM across all dataset, smoothing and radius combinations and find that it performs at least as good as the *maximum* dECE from a hyperparameter search of 2000 samples in 17/19 instances.

We use our results from the hyperparameter searches to run ADMM and dECE on a finer grid of certified confidences as referenced in section 5.2. Here, we present the results for the finer grid for CIFAR-10 (see Figure 15). The results for ImageNet are shown in Figure 3.

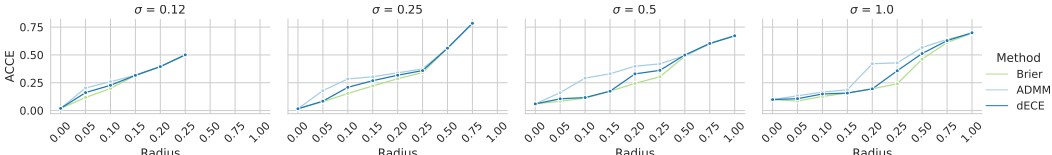

Figure 15: The ACCE is shown here for ADMM, dECE and the Brier confidences. One set of carefully selected hyperparameter is used. For a wider overview see Figure 13

## H   EXPERIMENTS - BRIER SCORE

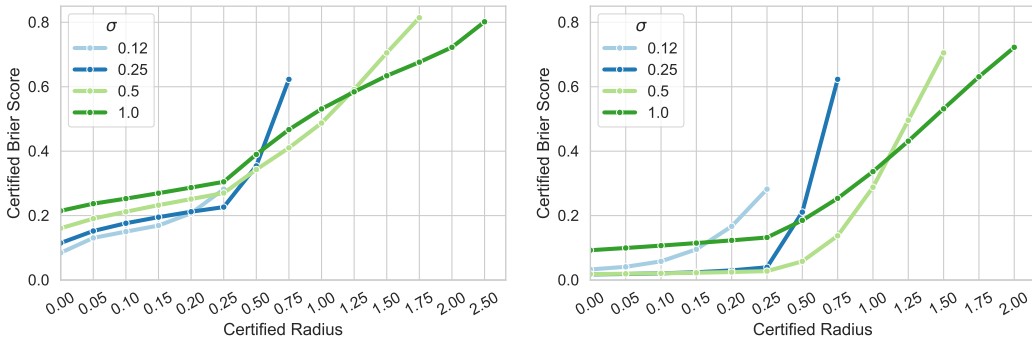

(a) Certified Brier scores (CBS) on CIFAR-10 across a range of certified radii for models with different smoothing $\sigma$. Each CBS is based on all certifiable samples at that radius.

(b) Certified Brier scores (CBS) on CIFAR-10 across a range of certified radii for models with different smoothing $\sigma$. Each data point is based on the same set data points per model.

In Figure 16a we plot the certified Brier scores on CIFAR-10 for a range of certified radii. Observe that the CBS increases significantly with certified radius and there is a trade-off

for CBS governed by $\sigma$: Heavily smoothed models achieve worse CBS on small radii but maintain it better for larger radii.

In 16b we plot the CBS on a constant subset of data that is certifiable at all shown radii. Hence, the certified accuracy is constant in this plot. We observe that these "most certifiable samples" maintain their CBS very well for small and medium radii. The CBS on these samples is significantly lower than the CBS on all certifiable samples as shown in 16a.

# I    Experiments - Adversarial Calibration Training

## I.1    ACCE Evaluation Parameters

In order to evaluate the ACCE during experiments on Adversarial Calibration Training, we report the maximum value over a grid of 16 runs. We do this to prevent overfitting the ACCE on a certain set of hyperparameters and thus ensure ACCE-ACT is not given an unfair advantage during evaluation. We run ADMM 16 times and pick the largest ACCE across them. The hyperparameters are given by the product of following parameters:

- Initialization of the confidences using the observed, clean confidences, and the Brier confidences.
- The learning rate for the confidences is 0.001 or 0.01.
- The learning rate for the binning variable is 0.01 or 0.1.
- The multiplicative factor for scheduling of the $\rho_i$'s for the ADMM solver: 1.004 or 1.01.

## I.2    Baseline Models

This section provides more details on how models from Salman et al. (2019) were chosen for fine-tuning.

**CIFAR-10.** In Table 14 and 15 in Salman et al. (2019), the authors list the parameters of their adversarial training runs and the certified accuracy per radius. For each certified radius we pick the best PGD result from each table (see Table 4) and perform certification on these checkpoints with 100,000 certification samples. We then use these certificates to obtain the ACCE and CBS, and verify that our certified accuracy lies within $\pm 3\%$ of Salman et al. (2019)'s reported results. Finally, we pick one or multiple models per certified radius on (or very close to) the Pareto curve between certified accuracy and ACCE, i.e., for each of the models there exist no other model that simultaneously improves ACCE and certified accuracy (also see Table 4). For all radii we pick the model achieving the highest accuracy and for larger radii we also pick two more models that are already better calibrated (at the cost of accuracy) to investigate whether our training can improve calibration on these models too. We use these models as baseline for fine-tuning.

**ImageNet.** For ImageNet, the authors list their models in Table 6 in Salman et al. (2019). For each certified radius we pick the best model. Analog to CIFAR10, we perform certification, verify that our results are close to Salman et al. (2019) and certify calibration. The results are shown in Table 5. We use each of these four models for fine-tuning.

Table 4: Certified metrics for the best models from Salman et al. (2019) on CIFAR-10. Results are based on 100,000 certification samples and the full CIFAR-10 test set. We report approximate certified calibration error (ACCE), certified accuracy (CA), the certified brier score (CBS) and whether that model was chosen as baseline for our fine-tuning experiments. The hyperparameters follow Salman et al. (2019) notation: $T$ is the number of adversarial steps, $m_{train}$ the number of Monte Carlo samples, $\sigma$ the smoothing parameter and $\epsilon$ is the $L_2$ upper bound on the adversary.

| $T$ | $m_{train}$ | $\sigma$ | $\epsilon$ | ACCE | | | | CA | | | | CBS | | | | Baseline for Radius |
|---|---|---|---|---|---|---|---|---|---|---|---|---|---|---|---|---|
| | | | | 0.05 | 0.20 | 0.50 | 1.00 | 0.05 | 0.20 | 0.50 | 1.00 | 0.05 | 0.20 | 0.50 | 1.00 | |
| 2 | 2 | 0.25 | 1.00 | 15.95 | 29.22 | 49.90 | - | 70.34 | 65.60 | 55.66 | 0.00 | 16.75 | 18.97 | 31.14 | - | - |
| 2 | 2 | 0.50 | 2.00 | 20.77 | 26.78 | 44.99 | 64.24 | 52.16 | 49.71 | 44.92 | 37.03 | 25.15 | 26.70 | 30.96 | 45.93 | - |
| 2 | 4 | 0.12 | 0.25 | 11.06 | 27.94 | - | - | 84.31 | 76.50 | 0.00 | 0.00 | 9.12 | 11.21 | - | - | 0.05, 0.20 |
| 2 | 4 | 0.25 | 0.50 | 17.85 | 21.39 | 42.89 | - | 76.00 | 70.16 | 55.71 | 0.00 | 12.64 | 13.88 | 21.21 | - | 0.50 |
| 2 | 4 | 1.00 | 2.00 | 24.60 | 26.36 | 36.52 | 50.66 | 44.93 | 42.44 | 38.00 | 30.87 | 28.32 | 29.55 | 32.39 | 37.46 | - |
| 2 | 8 | 0.12 | 0.50 | 12.09 | 30.43 | - | - | 83.44 | 76.67 | 0.00 | 0.00 | 9.54 | 13.56 | - | - | - |
| 10 | 1 | 0.12 | 0.50 | 17.14 | 35.30 | - | - | 77.76 | 71.34 | 0.00 | 0.00 | 12.84 | 18.71 | - | - | - |
| 10 | 1 | 0.50 | 0.50 | 18.26 | 31.00 | 38.13 | 53.38 | 62.31 | 58.07 | 48.17 | 32.64 | 18.50 | 20.41 | 23.34 | 32.17 | - |
| 10 | 1 | 0.50 | 1.00 | 15.98 | 31.13 | 44.16 | 59.81 | 57.27 | 54.32 | 47.06 | 35.10 | 22.04 | 23.93 | 28.16 | 41.36 | - |
| 10 | 1 | 0.50 | 2.00 | 19.93 | 24.94 | 45.73 | 64.35 | 48.48 | 46.08 | 41.42 | 33.89 | 25.74 | 27.11 | 31.18 | 46.18 | - |
| 10 | 2 | 0.50 | 0.50 | 17.98 | 30.26 | 33.59 | 50.97 | 64.11 | 59.08 | 49.40 | 32.62 | 17.06 | 18.61 | 20.94 | 28.67 | 0.50 |
| 10 | 2 | 0.50 | 1.00 | 16.21 | 30.61 | 39.51 | 56.36 | 61.02 | 57.43 | 49.50 | 36.10 | 19.66 | 21.40 | 24.97 | 36.25 | - |
| 10 | 2 | 0.50 | 2.00 | 22.34 | 29.03 | 45.76 | 64.59 | 52.59 | 50.16 | 44.88 | 37.20 | 25.92 | 27.38 | 31.53 | 46.52 | 1.00 |
| 10 | 2 | 1.00 | 2.00 | 22.70 | 24.18 | 35.11 | 46.71 | 43.10 | 41.01 | 36.90 | 30.40 | 27.53 | 28.62 | 31.05 | 36.13 | 1.00 |
| 10 | 4 | 0.12 | 0.25 | 11.32 | 28.50 | - | - | 83.41 | 75.96 | 0.00 | 0.00 | 9.63 | 12.06 | - | - | - |
| 10 | 4 | 0.25 | 1.00 | 15.21 | 27.25 | 49.30 | - | 72.13 | 67.24 | 56.98 | 0.00 | 15.45 | 17.52 | 28.96 | - | - |
| 10 | 4 | 0.50 | 0.25 | 19.29 | 27.93 | 29.87 | 44.96 | 66.23 | 60.10 | 47.07 | 26.89 | 15.04 | 16.40 | 16.79 | 22.00 | - |
| 10 | 4 | 0.50 | 1.00 | 16.28 | 29.62 | 37.69 | 52.78 | 63.46 | 58.87 | 49.54 | 34.30 | 18.97 | 20.57 | 23.51 | 32.90 | 1.00 |
| 10 | 8 | 0.12 | 0.25 | 10.90 | 27.83 | - | - | 84.00 | 75.78 | 0.00 | 0.00 | 8.70 | 10.94 | - | - | - |
| 10 | 8 | 0.25 | 0.50 | 17.20 | 20.53 | 41.34 | - | 76.78 | 70.45 | 55.20 | 0.00 | 12.01 | 12.94 | 19.51 | - | - |
| 10 | 8 | 0.25 | 1.00 | 15.21 | 28.68 | 49.65 | - | 73.43 | 68.58 | 57.62 | 0.00 | 15.22 | 17.25 | 28.88 | - | 0.50 |
| 10 | 8 | 1.00 | 2.00 | 26.14 | 28.13 | 41.08 | 50.96 | 46.36 | 44.03 | 39.32 | 31.35 | 28.36 | 29.52 | 32.34 | 37.40 | - |

Table 5: Certified metrics for the best four models from Salman et al. (2019) on ImageNet. Results are based on 100,000 certification samples and 500 test set samples. For notation refer to the caption of Table 4. For all models $T = 1$ and $m_{train} = 1$.

| $\sigma$ | $\epsilon$ | ACCE | | | | | CA | | | | | CBS | | | | |
|---|---|---|---|---|---|---|---|---|---|---|---|---|---|---|---|---|
| | | 0.05 | 0.50 | 1.00 | 2.00 | 3.00 | 0.05 | 0.50 | 1.00 | 2.00 | 3.00 | 0.05 | 0.50 | 1.00 | 2.00 | 3.00 |
| 0.25 | 0.50 | 15.81 | 50.18 | - | - | - | 66.00 | 58.00 | 0.00 | 0.00 | 0.00 | 16.65 | 32.23 | - | - | - |
| 0.50 | 1.00 | 15.46 | 36.30 | 54.24 | - | - | 57.60 | 52.80 | 45.60 | 0.00 | 0.00 | 17.82 | 23.40 | 35.16 | - | - |
| 1.00 | 2.00 | 13.19 | 32.53 | 37.97 | 56.78 | 80.93 | 42.80 | 38.60 | 35.80 | 28.80 | 21.80 | 19.72 | 22.78 | 27.14 | 40.29 | 66.25 |
| 1.00 | 4.00 | 16.05 | 28.47 | 38.67 | 56.38 | 82.57 | 37.60 | 34.00 | 30.40 | 24.40 | 20.40 | 21.05 | 23.22 | 26.91 | 40.91 | 68.88 |

## I.3 FINE-TUNING PARAMETERS

In this section, we provide more details on the hyperparameters for CIFAR-10 experiments. We fine-tune various model checkpoints provided by Salman et al. (2019). Generally, we adopt the number of Gaussian perturbations ($m_{train}$ in Salman et al. (2019) notation, $V$ in ours), the attack boundary $\epsilon$ and the smoothing $\sigma$ from the model checkpoint as trained before. We train using SGD with batch size 256 and weight decay of 0.0001. We use gradients obtained through backpropagation as recommended by (Salman et al., 2019) opposed to Stein gradients. We further follow the authors in setting the learning rate for the attack as function of $\epsilon$ and $T$, i.e. the attack bound and the number of optimization steps. However, we allow for some scaling and use a learning rate of $2v\frac{\epsilon}{T}$ with factor $v$ as additional hyperparameter.

**Brier-ACT.** We fine-tune for 10 epochs with a linear warm-up schedule for $\epsilon$ that reaches full size at epoch 3. We decrease the learning rate for the model weights every 4 epochs by a factor of 0.1. We train 16 models per baseline model that share these parameters, but form a grid over the factor to scale attack learning rates $v \in \{0.01, 0.1, 1.0, 2.0\}$, weight learning rates $\{0.01, 0.05\}$ and the number of steps used in the attack, $T \in \{1, 4\}$.

**ACCE-ACT.** We fine-tune for 10 epochs with $\epsilon$ schedule and weight learning rate schedule as described above. As the calibration loss is a distributional measure over a set of samples, a larger batch size is advisable and thus all of the runs are performed on a batch size of 2048. We train 16 models per baseline with hyperparameters sampled from the Cartesian product of the following hyperparameters: The attack learning rate scaling factor $v \in \{0.01, 0.1, 1.0, 2.0\}$, number of steps $T \in \{1, 4, 10, 20\}$, weight learning rates $\{0.01, 0.05\}$, and the factor in the multiplicative schedule of $\rho_i \in \{0.01, 0.05, 0.1, 0.5\}$ in the Lagrangian that governs how early the constraints will be enforced.

## I.4 CIFAR-10 FINE-TUNING EFFECTS

**CIFAR10 for all baseline models.** Above, we compare fine-tuning methods on the baseline models with the highest certified accuracy for each radius. However, due to the trade-off identified between certified accuracy and certified calibration, these models tend to have poor certified calibration. Therefore we also investigate whether, ACT is able to improve certified calibration on baselines that are naturally better calibrated. In Figure 17, you may observe that for each model that is certified for a radius $R \geq 0.5$, significant improvements are possible.

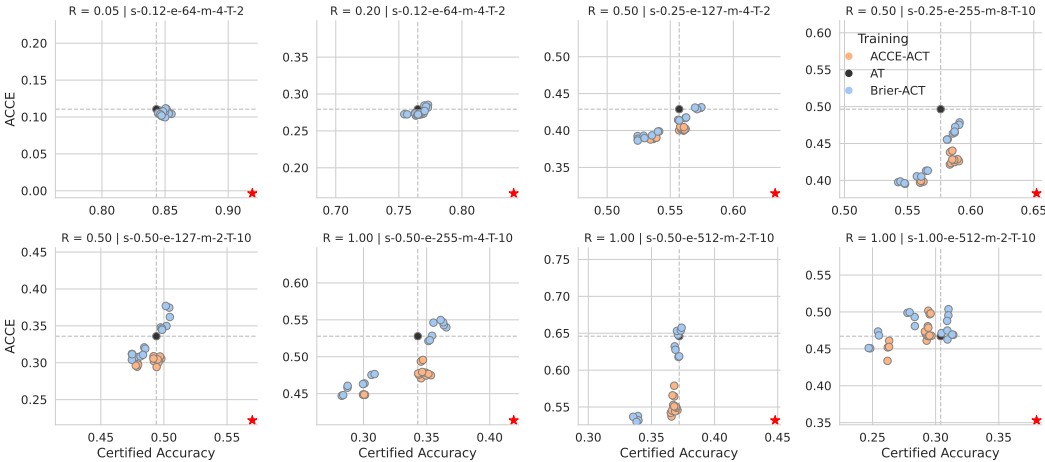

Figure 17: Each sub-figure depicts the effects of fine-tuning on the accuracy and ACCE for one pre-training model checkpoint. The pre-training model checkpoints ("AT") lie on the Pareto curve between ACCE and accuracy at the radius shown in the title. You may observe the ACCE-ACT yields significant ACCE improvements for all models at larger radii. These are achieved without adverse effects on the accuracy for all but the last model "R=1.00 | s-1.00-e-512-m-2-T-10".

## I.5 Ablation Studies

To investigate the effectiveness of ACT beyond further fine-tuning using standard AT, we also fine-tune the baseline models with AT. We use the same hyperparameters as for Brier-ACT as outlined in I.3. The results are shown in Table 6 (the setup is equivalent to Table 2). You may observe that fine-tuning the model using AT yields similar results to Brier-ACT but does not show the same effectiveness as ACCE-ACT. It is well-known that cross-entropy training is equivalent to maximum likelihood estimation. It is further known that the log likelihood is a proper scoring rule such as the Brier score. Hence, it is not surprising that careful utilisation for the cross-entropy loss is capable improving calibration. However, literature suggests that the Brier score can be more effective in training for calibration and cross-entropy tends to lead to overconfident predictions (Guo et al., 2017; Mukhoti et al., 2020; Hui and Belkin, 2021). In Figure 18, we show that fine-tuning ACCE-ACT outperforms AT on all tested baselines.

Table 6: Comparison of calibration for CIFAR-10 models achieving within 3% of the highest certified accuracy (CA) per training method across certified radii (from 0.05 to 1.00). Metrics are the approximate certified calibration error (ACCE ↓) and certified Brier score (CBS ↓).

| Metric | Method | 0.05 | 0.20 | 0.50 | 1.00 |
|---|---|---|---|---|---|
| CA | | [83, 86] | [74, 78] | [56, 59] | [35, 38] |
| ACCE | AT | 10.90 | 27.83 | 49.30 | 56.36 |
| ACCE | Finetuned AT | 9.99 | 27.15 | 42.72 | 51.71 |
| ACCE | Brier-ACT | **9.97** | **27.07** | **41.32** | 52.13 |
| ACCE | ACCE-ACT | 10.06 | 27.22 | 42.16 | **47.08** |
| CBS | AT | 8.70 | 10.94 | 28.88 | 36.25 |
| CBS | Finetuned AT | **8.09** | **10.12** | 20.86 | 31.89 |
| CBS | Brier-ACT | 8.10 | 10.20 | **19.38** | 32.54 |
| CBS | ACCE-ACT | 8.82 | 10.73 | 20.94 | **24.87** |

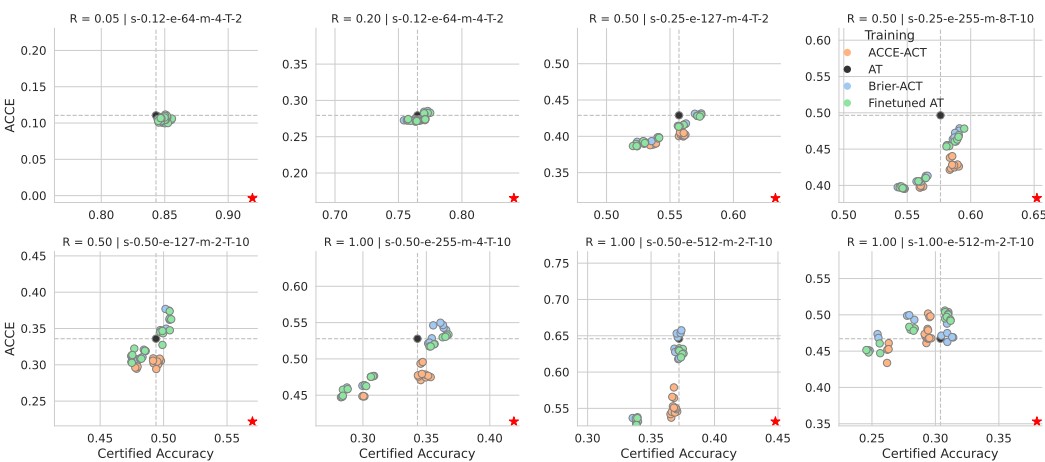

Figure 18: Fine-tuning results for every model checkpoint including Fine-tuning using AT.

## J Experiments - Adversarial Calibration Training - Further Datasets

### J.1 FashionMNIST Results

We replicate the CIFAR-10 experiments on FashionMNIST (Xiao et al., 2017) and find similar effects.

### J.1.1 EXPERIMENTAL SETUP

We replicate the CIFAR-10 experiments exactly, except for the following differences.

- We train our own baselines as Salman et al. (2019) do not discuss FashionMNIST. We use the hyperparameters they report for CIFAR-10, however, we use a linear warm-up schedule for $\sigma$ over 8 epochs. Further, we set $\epsilon = 0$ for 8 epochs and then use their warm-up schedule for 8 epochs. These adaptions were required to prevent models from diverging for $\sigma \geq 0.5$.
- We certify 500 samples on FashionMNIST rather than the full test-set on CIFAR-10 due to the cost of randomized smoothing.
- We additionally certify radius $R = 0.1$.
- We follow Zhai et al. (2020), who apply LeNet (LeCun et al., 1998) on MNIST and apply it to FashionMNIST.

### J.1.2 RESULTS

Below we present Table 7 for FashionMNIST, that replicates Table 2. We compare models that come within 3% of the model with the highest certified accuracy for each certified radius. We compare the best calibration achieved by any model. We observe that the Brier-ACT performs similar to AT (standard adversarial training). ACCE-ACT outperforms both other methods.

Table 7: Comparison of calibration for FashionMNIST models achieving within 3% of the highest certified accuracy (CA) per training method across certified radii (from 0.05 to 1.00). Metrics are the approximate certified calibration error (ACCE ↓) and certified Brier score (CBS ↓).

| Metric | Method | 0.05 | 0.10 | 0.20 | 0.50 | 1.00 |
|---|---|---|---|---|---|---|
| CA | | [87, 90] | [86, 89] | [82, 85] | [76, 79] | [60, 63] |
| ACCE | AT | 11.44 | 13.71 | 16.51 | 42.55 | 48.03 |
| ACCE | Brier-ACT | 10.96 | 12.96 | 15.20 | 42.35 | 48.03 |
| ACCE | ACCE-ACT | **10.32** | **11.26** | **13.19** | **39.62** | **43.78** |
| CBS | AT | 7.73 | 7.93 | 8.34 | 22.32 | 27.87 |
| CBS | Brier-ACT | 7.05 | 7.37 | 7.88 | 21.95 | 27.34 |
| CBS | ACCE-ACT | **6.76** | **7.07** | **7.28** | **17.77** | **21.51** |

## J.2 SVHN RESULTS

We replicate the CIFAR-10 experiments on the Street View House Number (SVHN) dataset (Netzer et al., 2011) and find similar effects.

### J.2.1 EXPERIMENTAL SETUP

We replicate the CIFAR-10 experiments exactly, except for the following differences.

- We train our own baselines as Salman et al. (2019) do not discuss SVHN. We use the hyperparameters they report for CIFAR-10, however, we use a linear warm-up schedule for $\sigma$ over 8 epochs. Further, we set $\epsilon = 0$ for 8 epochs and then use their warm-up schedule for 8 epochs. These adaptions were required to prevent models from diverging for $\sigma \geq 0.5$. The baselines perform similar to (Zhai et al., 2020).
- We certify 500 samples on SVHN rather than the full test-set on CIFAR-10 due to the cost of randomized smoothing.
- We additionally certify radius $R = 0.1$.

### J.2.2 RESULTS

Below we present Table 8 for SVHN, that replicates Table 2. We compare models that come within 3% of the model with the highest certified accuracy for each certified radius. We

compare the best calibration achieved by any model. We observe that the Brier-ACT performs better than AT (standard adversarial training) on small radii. ACCE-ACT outperforms both other methods across all conditions.

Table 8: Comparison of calibration for SVHN models achieving within 3% of the highest certified accuracy (CA) per training method across certified radii (from 0.05 to 1.00). Metrics are the approximate certified calibration error (ACCE ↓) and certified Brier score (CBS ↓).

| Metric | Method | 0.05 | 0.10 | 0.20 | 0.50 | 1.00 |
|--------|--------|------|------|------|------|------|
| CA | | [89, 92] | [87, 90] | [81, 84] | [61, 64] | [27, 30] |
| ACCE | AT | 14.31 | 15.04 | 18.42 | 41.09 | 60.18 |
| ACCE | Brier-ACT | 11.55 | 12.63 | 15.09 | 44.37 | 61.98 |
| ACCE | ACCE-ACT | **11.40** | **12.15** | **14.20** | **39.29** | **49.68** |
| CBS | AT | 6.79 | 6.53 | 7.40 | 18.25 | 41.64 |
| CBS | Brier-ACT | 4.73 | 5.08 | 5.58 | 18.56 | 44.26 |
| CBS | ACCE-ACT | **4.49** | **4.51** | **5.45** | **16.21** | **26.08** |

## J.3 CIFAR-100 Results

We replicate the CIFAR-10 experiments on CIFAR-100 (Krizhevsky, 2009) and find similar effects.

### J.3.1 Experimental Setup

We replicate the CIFAR-10 experiments exactly, except for the following differences.

- We train our own baselines as Salman et al. (2019) do not discuss CIFAR-100. We use the hyperparameters they report for CIFAR-10, however, we use a linear warm-up schedule for $\sigma$ over 8 epochs. Further, we set $\epsilon = 0$ for 8 epochs and then use their warm-up schedule for 8 epochs. These adaptions were required to prevent models from diverging for $\sigma \geq 0.5$.

- We certify 500 samples on CIFAR-100 rather than the full test-set on CIFAR-10 due to the cost of randomized smoothing.

- We additionally certify radius $R = 0.1$.

### J.3.2 Results

Below we present Table 9 for CIFAR-100, that replicates Table 2. We compare models that come within 3% of the model with the highest certified accuracy for each certified radius. We compare the best calibration achieved by any model. We observe that the Brier-ACT performs best for small radii and ACCE-ACT for large radii. Both outperform the AT baseline.

Table 9: Comparison of calibration for CIFAR-100 models achieving within 3% of the highest certified accuracy (CA) per training method across certified radii (from 0.05 to 1.00). Metrics are the approximate certified calibration error (ACCE ↓) and certified Brier score (CBS ↓).

| Metric | Method | 0.05 | 0.10 | 0.20 | 0.50 | 1.00 |
|--------|--------|------|------|------|------|------|
| CA | | [45, 48] | [43, 46] | [38, 41] | [28, 31] | [15, 18] |
| ACCE | AT | 29.00 | 32.23 | 35.52 | 55.77 | 60.46 |
| ACCE | Brier-ACT | **25.14** | 31.54 | 32.91 | 56.89 | 55.88 |
| ACCE | ACCE-ACT | 27.21 | 30.70 | **30.95** | **51.85** | **53.11** |
| CBS | AT | 19.08 | 20.77 | 23.86 | 37.80 | 43.71 |
| CBS | Brier-ACT | **18.09** | **19.93** | **21.13** | 39.60 | 36.99 |
| CBS | ACCE-ACT | 18.78 | 20.33 | 22.16 | **34.35** | **32.04** |

## J.4 IMAGENET RESULTS

In this section, we provide details for the adversarial training experiments we conducted on ImageNet (Deng et al., 2009).

### J.4.1 EXPERIMENTAL SETUP

For ImageNet we pick the best PGD model for each certified radius reported by Salman et al. (2019) in their Table 6. These baselines and their certified metrics can be found in 5. For these four models, we perform fine-tuning with Brier-ACT and ACCE-ACT. The experimental setup is similar to that for CIFAR-10 (see Section 5.3). We train with batch size 256 for Brier and 1024 for ACCE, and fine-tune 6 models per baseline. The hyperparameters for these 6 models are the product of weight learning rates $\{0.001, 0.005, 0.01\}$ and attack learning rate factor of $\{1.0, 5.0\}$ (see I.3 for an explanation). For all ACCE-ACT models we use the same ADMM parameters ($\rho = 0.01$). Due to the cost of certification, only use 10'000 Gaussian noise samples to obtain the smooth model.

### J.4.2 RESULTS

In Table 10, we compare models that come within 3% of the model with the highest certified accuracy for each certified radius. We compare the best calibration achieved by any model per training method. The calibration of ACT is consistently, but only marginally below that of AT. The best performing method here is "ACCE-ACT*", which is a variant of ACCE-ACT in which we update model weights using loss $\eta\mathcal{L}_{CE}(\bar{f}(\mathbf{x}+\gamma^*), y)+(1-\eta)ACCE(\bar{f}(\mathbf{x}+\gamma^*), a^*, y)$. We pick $\eta$ between 0.8 and 0.99, as the ACCE may be magnitudes larger than the CE. We think using the ACCE as part of the loss for updating the weights is a promising direction, but leave this for future work.

Table 10: Comparison of calibration for ImageNet models achieving within 3% of the highest certified accuracy (CA) per training method across certified radii (from 0.05 to 1.00). Metrics are the approximate certified calibration error (ACCE ↓) and certified Brier score (CBS ↓).

| Metric | Method | 0.05 | 0.50 | 1.00 | 2.00 |
|--------|--------|------|------|------|------|
| CA | | [66, 69] | [57, 60] | [44, 47] | [27, 30] |
| ACCE | AT | 23.08 | 63.69 | 66.09 | 68.33 |
| ACCE | Brier-ACT | **21.60** | 62.85 | 64.29 | 68.58 |
| ACCE | ACCE-ACT | 22.45 | 62.62 | 64.99 | 67.92 |
| ACCE | ACCE-ACT* | 21.67 | **61.96** | **64.23** | **67.56** |
| CBS | AT | 17.20 | 42.95 | 45.84 | 49.71 |
| CBS | Brier-ACT | 16.45 | 41.69 | **43.48** | 49.01 |
| CBS | ACCE-ACT | 16.79 | 41.73 | 44.41 | 48.17 |
| CBS | ACCE-ACT* | **16.28** | **40.84** | 43.75 | **47.91** |

## K ROBUSTNESS-CALIBRATION-TRADE-OFF

Previous literature on accuracy certification identifies a robustness-accuracy trade-off governed by a smoothing parameter $\sigma$: Strongly smoothed models achieve high certified accuracy on large radii, but tend to perform worse within small radii and vice versa (Cohen et al., 2019; Salman et al., 2019). We observe a similar trade-off for certified calibration: Models with small $\sigma$ provide tighter certified calibration on small radii, while heavily smoothed models maintain certified calibration much better. This becomes apparent for both the ACCE (see Figure 15) and CBS (see Figure 16a).

Further, we observe an association between calibration (ACCE and CBS), certified accuracy (CA) and average certified radius (ACR) (Zhai et al., 2020) that is heavily moderated by the certification radius. In Figure 19, the rank correlation between these metrics across all baseline and fine-tuned models are shown. For small radii, we observe that metrics *harmonise*: Large CA is associated with low ACCE and CBS. As the certification radius increases, these relationships invert, and we observe metrics *competing*: High CA is associated with high CBS and ACCE, i.e., poor calibration and low ACR is associated with poor calibration.

These relationships imply that there is a trade-off between calibration and accuracies at large certified radii. Certified accuracy and ACR both benefit strongly from large margins between the top and the runner-up confidence, which can increase overconfidence and thus harm calibration. We leave a closer investigation of the causes for future work.

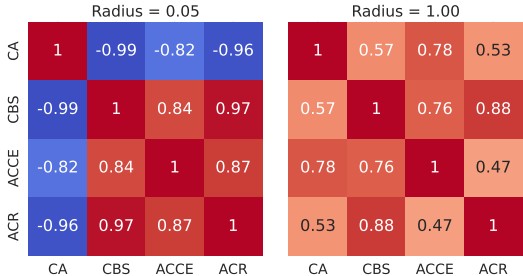

Figure 19: Spearman rank correlation for *certified accuracy* (CA ↑), *Certified Brier Score* (CBS ↓), *Approximate Certified Calibration Error* (ACCE ↓) and *average certified radius* (ARC ↑) for different certified radii. For small radii, optimal CBS, CA, and ACCE coincide. For larger radii there is a trade-off.

## L    Future Work

Our study offers an extensive analysis of certified calibration while highlighting avenues for future research. Initially, we note the capability of the MIP to optimise adaptive binning ECE, which we do not explore due to limitations of the baseline against which we compare. Additionally, our ACCE method approximates an upper limit for the *estimated* ECE, paving the way for applying confidence intervals to better align this metric with the true ECE (e.g., Kumar et al. (2019)). Further, we suggest to perform adversarial training as an inverse problem, where the targets are given by the worst-case confidences on the dataset. We also recognize a calibration generalization gap in ImageNet, suggesting that a meta-learning framework might yield improvements (Bohdal et al., 2023).

## M    Differentiable Calibration Error

### M.1    Definition

Bohdal et al. (2023) propose a differentiable approximation to the ECE estimator. The authors note that the standard calibration error estimator is non-differentiable in two operations: the accuracy and the hard binning. As we are assuming certified predictions, their correctness $c_n$ (and thus the accuracy per bin) is constant with respect to the confidence scores $z_n$ and thus does not need to be differentiable. We therefore simplify their dECE and apply only one differentiable approximation. We restate the calibration error estimator from (1) and simplify:

$$\hat{\mathrm{ECE}} = \sum_{m=1}^{M} \frac{|B_m|}{N} \left| \frac{1}{|B_m|} \sum_{n \in B_m} c_n - \frac{1}{|B_m|} \sum_{n \in B_m} z_n \right| \tag{32}$$

$$= \frac{1}{N} \sum_{m=1}^{M} \left| \sum_{n=1}^{N} a_{n,m}(c_n - z_n) \right| \tag{33}$$

where $a_{n,m} \in \{0,1\}$ is the hard indicator whether data point $n$ is assigned to bin $m$. This is replaced with a soft indicator $s_{n,m} \in [0,1]$ with $\sum_m s_{n,m} = 1$. Define matrix $\mathbf{S} \in [0,1]^{N \times M}$ with elements $s_{n,m}$ and let $\mathbf{e} = [c_1 - z_1, \ldots, c_N - z_N]$. We can write the differentiable calibration error $\hat{\mathrm{dECE}}$ as:

$$\hat{\mathrm{dECE}} = \frac{1}{N} \|\mathbf{S}^{\top} \mathbf{e}\|_1 \tag{34}$$

The soft assignment $s_{n,m}$ is obtained the following way. For $M$ equal width bins with cut-offs $\beta_1 < \ldots < \beta_{M-1}$, we define $\mathbf{b}$ with elements $b_i = -\sum_{m=1}^{i-1} \beta_m$. Further, let $\mathbf{w} = [1, 2, \ldots, M]$ we obtain the soft assignments $\mathbf{s}_n$ through a tempered softmax function: $\mathbf{s}_n = \boldsymbol{\sigma}((\mathbf{w}z_n + \mathbf{b})/\tau)$. For $\tau \to 0$, the vector $\mathbf{s}_n$ approximates a one-hot encoded vector and (34) recovers the original ECE.

## M.2  dECE Parameters

As for ADMM, we perform an extensive hyperparameter search for the dECE. Our key insight is to start the optimization with high values of $\tau$, i.e. 0.01 as this increases the smoothness of the objective function and slowly decrease $\tau$ to about $1 \times 10^{-6}$ at which point we usually observe a difference between the ECE and dECE of less than $1 \times 10^{-6}$.

We use the dECE as one method to approximate the CCE. As for ADMM, we test a wide range of hyperparameters for the dECE. The one most significant hyperparameter is the initialization of $\mathbf{z}$. We test the same initialization as for ADMM, i.e. the adversary-free and Brier confidences. In addition, we test random Gaussian and random uniform initialization. We do not find a single optimal strategy. It is possible that the adversary-free ECE is *higher* than the ECE obtained by using the Brier confidences and hence the Brier confidences are not universally better values to initialize $\mathbf{z}$. As dECE shows poor performance in exploring the loss surface, we recommend initialization to whatever initialization yields the largest error to begin with. All other hyperparameters are insignificant in comparison and usually over 95% of variation in results explained by the initialization of $\mathbf{z}$. We test various learning rates, schedulers for $\tau$ (as mentioned above), learning rate schedulers (Cosine Annealing, Constant, ReduceOnPleateau) and optimizer momentum and find none of these hyperparameters to be significant.

## N  Compute Cost

We outline the computational cost required to replicate experiments. We mostly use A40 GPUs and equivalent older models.

- The experiments in the motivation section require adversarial inference and possibly training. GPU hours 70 including training.
- The experiments performing parameter search for ADMM and dECE to estimate the ACCE take around 400 GPU hours.
- Training baselines that are not provided by Salman et al. (2019) requires 600 GPU hours.
- Performing randomized smoothing on baseline models takes 1000 GPU hours.
- Performing randomized smoothing on fine-tuned models takes 2200 GPU hours.
- Fine-tuning models takes 350 GPU hours.
- Running an ensemble of ADMMs to estimate the ACCE for the evaluation of baselines requires 320 GPU hours (baselines are certified on many radii).
- Running an ensemble of ADMMs to estimate the ACCE for fine-tuned models requires 90 GPU hours.

In total, we use around 5000 GPU hours for the experiments in this paper; additional exploratory experiments are not included.

