# OpenReview forum: "Towards Certification of Uncertainty Calibration under Adversarial Attacks"
_ICLR.cc/2025/Conference — ICLR 2025 Poster_

### Official Review · Reviewer_oWYq · 2024-10-28

**Soundness:** 2
**Presentation:** 3
**Contribution:** 2
**Rating:** 5
**Confidence:** 4

**Summary:**

The authors consider the problem of constructing certifications for neural networks in parallel with the task of improving confidence. The authors achieve this through the development of a new approximate certification measure, and support their work with significant experimentation.

**Strengths:**

The authors investigate an interesting extension of the certification literature, which moves beyond more traditional (and incremental) advances in the field of image based certifications.

In doing this, the authors introduce new techniques and approaches, which are novel and interesting, and the mathematical presentation is well detailed (although I have issues regarding specificity and rigour in explanations, see below).

**Weaknesses:**

I hold a number of concerns regarding this paper. One of the primary issues relates to this technique relying on an approximation to construct ACCE. I'm intrinsically cautious about any result that presents itself as a certification while relying on breaking the guarantees of the certification in order to calculate results. Expectations around certifications guide their use, and violating the precepts of a certification while still labelling it as a certification is problematic to me. To me,  an approximation to a guarantee is only useful in rare circumstances, where other insights can be gained by the approximation process - something that I would argue is not present within this work.

When it comes to the motivation for their work, the authors discuss possible reasons for an attack on calibration, however I would suggest that the current presentation of this is a bit sparse - it is sufficiently detailed for comparisons, but the ultimate applicability of this attack could require more justification. This perception is reinforced by statements being made regarding risk and processes that might support this work without citations - leaving this reader curious as to if the justifications actually exist.

While the sentence-by-sentence writing is overall well done (although a touch clunky at times), I would argue that the ordering of content is somewhat jumbled, and it hinders readability and accessibility. For example, certifications are required knowledge to understand content long before they are formally introduced on Line 177. The document, as structured, has multiple examples of such structural issues that would make it difficult for a reader to parse content if they were unfamiliar with the space.

These issues with presentation extend to how some of the results are presented. Take Table 1 as an example - what  is the reader meant to get out of this? If I understand it correct the Unattacked rows correspond to the base values, and ideally the AdaECE should decrease in the case of a successful attack? But when there is the switching behaviour about $\eta$, interpreting this table is nigh on impossible. Moreover, the compressed nature of the table, and its positioning within the document further hinder accessibility.

General comments regarding identified issues follow below:
- L118 "has been discussed" -> "have been discussed". Moreover, in this sentence you use "label attacks" without explaining what these are, nor what an adversarial attack even is.
- L119 "their immediate effects" - what is the subject of their: adversarial attacks? The literature? Label attacks?
- L119 "effects on the reliability of uncertainty estimation are often overlooked" you can affect somethings reliability, but you cant affect the reliability. The reliability is a downstream consequence of the impact made.
- L120 after the colon - why is this colon required? And why are these things overlooked if this is a statement that can be made without citation, suggesting it is so well known that such a justification is unnecessary.
- L121 "Beyond label attacks" - the subject of this paragraph, to this point, has been label attacks. If a different framework is to be introduced, this should be a new paragraph. Is the point here not that it's not only that attacks can indirectly influence confidence, motivated attackers can attempt to directly manipulate the confidence scores as well?
- L125 "We note that this impacts system-level uncertainty and conclude that both label and confidence attacks leave system calibration vulnerable" - you're noting the very things you've just stated in the preceding sentences, without stating why this is important.
- Line 127 "for both, their" - comma is unnecessary
- Line 128  "when uncertainty estimates...." - this sentence fragment is disconnected from its context. Is it not "machine learning systems incorporating uncertainty estimates in their decision processes are monitored regularly for both their .... in safety-critical applications." ?
- Line 127-129 - this is a statement of fact, are there any citations available for it?
- Line 132 Denial of Service should be capitalised appropriately, as it is the name of a thing.
- Line 120 and line 132 - incorrect use of colons. For one, they're unnecessary in these contexts, and for two they shouldn't have capitalisation after the colon (although there's some nuance here between British and American English, still, fundamentally, the colons are unnecessary).
- Line 132: The defences are again a whole new concept, should this not be a new paragraph for readability?
- Line 137: Discussion centred around certified models without properly defining what these are? Readers who are not familiar with the certification literature will have limited opportunities to catch up up here.
- Table 1 - font too small, it's what, half the size of the standard maths font, and is really jammed in to the paper.
- Line 195 "The certificate on the prediction" - is the first the necessary?. And what does "showing a certificate on the confidence" mean.
- "extended by Salman et al. (2019) and Kumar et al. (2020) showing a certificate on the confidence" - this implies that both Salman AND Kumar certified confidence, rather than just Kumar.

**Questions:**

Answers to the following questions from the authors would be appreciated:
- Could you comment upon how you view the process of calculating/employing an approximate certification?
- Which norm (see, for example, line 179-188) are you using?
- Is there any sensitivity to the number of bins (M), or their partitioning?
- Are there any uncertainties that could be calculated on Figure 3 or Table 2? For table 2 how did you settle upon selecting "within 3% of the ihghest certified accuracy"
- In Figure 4, you claim "ACCE-ACT significantly improves certified calibration at larger radii" - why is this not present at smaller radii, and how much is this actually an improvement? The overlapping nature of the circles makes it very difficult to parse out the exact impact here, beyond tea-leaf reading.

---

> ### Author Response · Authors · 2024-11-25
> **Rebuttal 1**
>
> We thank the reviewer for taking the time to read our work, providing such detailed feedback and recognising the novelty of our work. We are pleased they find the work interesting and our presentation well detailed.
>
> We thank the reviewer for raising minor issues on language. We will not comment individually here, but appreciate the feedback and incoporate them into our revision.
>
> > W1: [...] this technique relying on an approximation to construct ACCE. I'm intrinsically cautious about any result that presents itself as a certification while relying on breaking the guarantees of the certification in order to calculate results. Expectations around certifications guide their use, and violating the precepts of a certification while still labelling it as a certification is problematic to me. To me, an approximation to a guarantee is only useful in rare circumstances, where other insights can be gained by the approximation process - something that I would argue is not present within this work.
>
> The reviewer is correct in noting that the ACCE is indeed approximate, which is a limitation of the work presented. We would like to note that we are very transparent about it, naming the metric *approximate*.
>
> Further, in this work we propose multiple metrics. The certified Brier score (CBS) is available in closed-form and an exact bound (see Eq 3). Second, we show that the certified calibration error can be seen as a mixed-integer program (see Eq 6). The solution to this MIP is an exact bound. The approximate nature of the ACCE comes from solving this using ADMM.
>
> We believe this inexact solution to the MIP is still very useful.
> 1. The ACCE provides a pessimistic view on the calibration error under attack. Certificates are often used in an apriori risk assessment of the model under deployment. While the ACCE cannot provide an exact guarantee, we argue that it still provides *useful* information on possible model failures. The practitioner should, of course be weary to make a clear distinction between an exact and an approximate solution.
> 2. Many certication methods such as Gaussian smoothing provide non-deterministic certificates (Cohen et al 2019). Hence, with small probability ($< \alpha$) the certificate is false. As noted by the reviewer, our ADMM solution to the MIP does not enjoy such exact bounds on falseness. However, we believe this shows that the community is generally interested in fallable certificates, if they are useful. Further, we should note that many certification techniques provide an exact guarantee for the accuracy *on a specific testset* failing to establish exact certificates on metrics w.r.t. the samples presented during deployment.
> 3. The inexact nature allows us to perform adversarial training as we present with our ACT-ACCE method. Performing certified training has been vital in the field to improve certicates.
>
> > W2: [...] motivation for their work [...] is a bit sparse - it is sufficiently detailed for comparisons, but the ultimate applicability of this attack could require more justification. This perception is reinforced by statements being made regarding risk and processes that might support this work without citations - leaving this reader curious as to if the justifications actually exist.
>
> Before diving into motivation of attacking uncertainty, we would like to reiterate that *any adversarial attack* will yield changes in calibration. If a system uses uncertainty estimates, adversarial attacks will impact the reliability of the system. That includes the classical attack maximizing the cross-entropy. This should provide considerable motivation to many practitioners to pay more attention to uncertainty and not just accuracy when attacks are expected.
>
> Now towards uncertainty attacks: We are not the first to argue that attacks on uncertainty estimates are possible. Galil & El-Yanif (2021) argue that attacks on individual confidence scores (i.e. sample-level uncertainty estimates) are plausible and motivate this extensively. Kumar et al. (2020), who present the certification of confidence, conclude their paper stating that "However, this method does not produce any guarantees on the calibration of the underlying model itself." recognising the need for another level of certification for calibration of the model, which we provide.
>
> As we are the first to show direct attacks on the model reliability, there is no literature that we can cite on their reception. We believe that these attacks are conceivable and will eventually be carried out.
>
> Regarding our statements on the risk: Any decision system has associated risks that clearly change when the decision model is misspecified (e.g. due to adversarial attacks). In the credit lending example, credit risk management is a core discipline of finance. We gladly elaborate and cite literature on this. Could the reviewer clarify where they do not find our arguments convincing.

---

> > ### Author Response · Authors · 2024-11-25
> > **Rebuttal 2**
> >
> > > W3: While the sentence-by-sentence writing is overall well done [...] the ordering of content is somewhat jumbled. For example, certifications are required knowledge to understand content long before they are formally introduced on Line 177. [...].
> >
> > We thank the reviewer for this fedback. We mention what certification is early on (L39-L42), but only formally introduce it much later (Section 3 L174-L201). In Appendix A.1, we give a primer on calibration. We reference this earlier now to provide earlier guidance. We hope that increases readability for readers unfamiliar with certification.
> >
> > Overall, we are aware that the paper is very dense. We need to introduce two subfields (calibration and certification) for readers who are unfamiliar with one of them. We then motivate the problem carefully, followed by numerous contributions (proposing the certification framework, deriving the CBS, derive the MIP for the CCE, approximating it using the ACCE, and adversarial training). Doing this in 10 pages inadvertantely leaves us with compromises. Nonetheless, we think the paper is well structured and reviewers Xy4o and 2ZV2 have explicitly stated they found it easy to follow.
> >
> > If the reviewer is aware of more concrete examples, where they find concepts are not timely introduced, we gladly will incorporate that.
> >
> > > W4a: These issues with presentation extend to how some of the results are presented.
> >
> > With the introduction of certified calibration, evaluating models becomes a multi-objective problem: accuracy vs robustness vs calibration. Our presentation of results must reflect that. That is, e.g., why we take two different strategy of presenting the training results, Table 2 and Figure 4. Table 2 fixes a range of accuracies across models, while Figure 4 centers on post-training effects for certified acucracy and ACCE. Apart from the Pareto front there is no clear ordering or weighing of importance of accuracy vs ACCE.
> >
> > > W4b: Take Table 1 as an example - what is the reader meant to get out of this? [...] when there is the switching behaviour about $\eta$, interpreting this table is nigh on impossible. [...] the compressed nature of the table, and its positioning [...] hinder accessibility.
> >
> > Table 1: The purpose is to demonstrate that calibration can easliy be attacked without the accuracy ever changing. Hence, monitoring accuracy does not sufficiently protect calibration. Therefore, we present the baseline calibration error ("Unattacked") and show that some parameterization of our attack can yield in *increased* AdaECE (higher AdaECE is worse).
> >
> > The important bit is that it is possible to worsen calibration; which $\eta$ is not important for the progression of the paper. Nonetheless, we are happy to clarify. Consider Figure 6, which shows reliability diagrams for these attacks. Figure 6a shows that the classifier is slightly overconfident. When attacking with $\eta=-1$ and $\omega=\hat{y}$ our objective minimises the loss w.r.t. to the predictions. This pushes the softmax scores to 1 for all observations. All combinations of $\eta, \omega$ are shown in Figure 6 and should help illustrate what each optimization achieves.

---

> > > ### Author Response · Authors · 2024-11-25
> > > **Rebuttal 3**
> > >
> > > > Q1: Could you comment upon how you view the process of calculating/employing an approximate certification?
> > >
> > > See our response to W1.
> > >
> > > > Q2: Which norm (see, for example, line 179-188) are you using?
> > >
> > > We are not refering to any norm in particular. If certificates **C1** and **C2** can be proven for any norm on $\gamma$ or more generally any set of inputs $\mathbf{x}_n$, our method is applicable and calibration is certified w.r.t. that same set. We clarify this in the revised version.
> > >
> > > > Q3: Is there any sensitivity to the number of bins (M), or their partitioning?
> > >
> > > We thank the reviewer for this interesting question. We find that the certified Brier score (CBS), which does not rely on binning, is correlated highly with the ACCE (r=0.98) indicating that binning is not an artifact confounding the ACCE. Nonetheless, the canonical estimator for the ECE (see Eq 2) depends on the binning and hence our metrics inherits this.
> > >
> > > > Q4a: Are there any uncertainties that could be calculated on Figure 3 or Table 2?
> > >
> > > In the context of worst-case estimation, we find that mean+std aggregation to reflect uncertainties is often not very informative. Instead, we replicate Figure 3 in Figures 13 and 14 showing the $max$ over a large number of hyperparameters. For Table 2, we provide Figure 4 which shows the effectiveness of ACT from a different angle and presents raw data points rather than summary statistics.
> > >
> > > > Q4b: For table 2 how did you settle upon selecting "within 3% of the ihghest certified accuracy"
> > >
> > > Model selection is a multi-objective problem when regarding calibration and accuracy (as argued for W4a). In Table 2, we only regard the most accurate models. 3\% of loss in accuracy seemed like a reasonable degredating in accuracy that practitioners might be willing to sacifice for better calibration. However, this is *highyly* dependent on the application. To show that the choice of X\% does not impact the interpretation of the results, we present Figure 4 with raw data to show that ACT can be effective in improving certified calibration.
> > >
> > > > Q5: In Figure 4, you claim "ACCE-ACT significantly improves certified calibration at larger radii" - why is this not present at smaller radii, and how much is this actually an improvement? [...].
> > >
> > > In this Figure, we present the changes in the ACCE, while norming the ACCE to [0,1]. For instance, at R=1, the best improvement in calibration without major cost to accuracy is the orange circle at (-0.01, -0.11). This means: at only 0.01 cost to accuracy, the ACCE improved by 0.11.
> > >
> > > The question about the lack of effect for smaller circles in Figure 4 is interesting. First, while this very much is a trend, it is not uniformly true. For example, see Table 7 for SVHN where we get a change from 6.79 to 4.49 for small radii (R=0.05), which is a 33% improvement. We do not know why this is the case, but would be very interested to read about it in future work.

---

> ### Comment · Reviewer_oWYq · 2024-11-25
>
> Thank you to the authors for your detailed response - I was starting to wonder if my review had been ignored due to the time delta in responses, but clearly you've put work into this, and that's appreciated. This message is primarily just to note that I've read your response and I'm digesting it, and I will get back to you if I have any additional questions.
>
> I will stress, however, that while the updates to the paper are appreciated, I hold significant reticence relating to changing your score due to the presentation of approximation certifications. I know we disagree on this point, and that this is an unappreciated comment, however in my eyes, labelling anything as a certification, even with a qualifying statement like calling it approximate, conveys a level of expectation in a user. Because fundamentally, it is still being read as a guarantee.

---

> > ### Author Response · Authors · 2024-11-28
> >
> > We thank the reviewer for their thoughtful engagement and detailed comments. While we disagree on this, the concerns regarding the terminology and the expectations it may set for users are noted and appreciated.
> >
> > Could the reviewer comment on whether they consider a metric such as the ACCE—independent of its specific terminology — useful in the context of this work.

---

> ### Comment · Reviewer_oWYq · 2024-12-01
>
> Apologies for taking a few days to get back to you on this (and for this discussion happening so late into the discussion period), but I wanted to make sure I wasn't missing any potential applications due to the nature of my background in this space.
>
> In all honesty: I do not see a context where a certification is required but where an approximation like ACCE could be useful. Certificates are designed for safety critical applications, and approximate-safety means nothing if we are in a position to require it. And in your case, this is an approximation without even a guide as to how tight it is.
>
> To flip this around - under what conditions do you as the authors see ACCE as being usable? What benefit would it have to the community? And if you were deploying a certification in a context where it was necessary, under what conditions would you prefer to use an approximation rather than expending extra compute for the whole thing?

---

> ### Author Response · Authors · 2024-12-04
>
> We thank the reviewer for their continued engagement in the review process and appreciate the discussion. While we disagree on the utility of the ACCE at this point, it seems that we have addressed the reviewers remaining concerns.
>
> > In all honesty: I do not see a context where a certification is required but where an approximation like ACCE could be useful. Certificates are designed for safety critical applications, and approximate-safety means nothing if we are in a position to require it. And in your case, this is an approximation without even a guide as to how tight it is.
>
> We agree that an approximation will fall short "if we are in a position to require" an exact certificate on the ECE, but we believe, the ACCE is still significantly more informative than not having it. If certified calibration in general is sufficient, the CBS covers that.
>
> > To flip this around - under what conditions do you as the authors see ACCE as being usable? What benefit would it have to the community? And if you were deploying a certification in a context where it was necessary, under what conditions would you prefer to use an approximation rather than expending extra compute for the whole thing?
>
> We will outline some benefits the ACCE brings to the research community and practitioners.
>
> ### Research Community
> * To reiterate, the MIP in (6) with the ACCE solution obtained through ADMM is useful to augment training as we propose with the MIP in (8).
> * We are the first to study protecting the ECE and believe our work marks a major step forward.
> * Optimisation of the expected calibration error has been studied in recent years. For instance, the $dECE$, $MMCE$ and $ECE^{KDE}$ have been proposed (Bohdal 2023, Popordanoska 2022, Kumar 2018; in our preliminary experiments $dECE$ uniformly outperformed the others.). The ACCE at convergence is an exact expected calibration error estimator (see Figures 10 and 11) and hence our work marks a major step forward in the literature on optimisation of calibration properties, in our case the actual ECE rather than some approximation.
> * We believe the MIP in (6) provides the basis for others to employ better optimization techniques to extend it / solve it.
> * Our ACCE is a sets a new SOTA for calibration attacks.
>
> ### Practitioners
> We assume, a practitioner want deploys a model and is concerned about accuracy and uncertainty calibration under attack. The practitioner would like to estimate the apriori risk under attack. Hence, the practitioner chooses to use a model with smoothness guarantees and certifies accuracy.
>
> **Without this work.** They would be lost to judge how poor calibration can be. They might use standard $\max \mathcal{L}_{CE}$ attacks on the smooth model to observe a) Brier score and b) expected calibration error.
>
> **With this work.** They observe the certified Brier score (CBS) in closed form as exact bound providing clear apriori understanding of miscalibration under attack. However, this does not tell them how poor the *expected calibration error* (ECE) metric can become. In the absence of an exact certificate on the calibration error, the strongest empirical calibration error, i.e. ACCE, is very informative providing apriori information on how poor the ECE can become. If the ACCE is large and the deployer decides that the associated risks are too high, the deployer might consider not deploying the model, having gained valuable information from the ACCE. Contrary, should the ACCE be sufficiently small, the practitioner reaches a decision based on CBS and ACCE, knowing that a) miscalibration is truly bounded through the CBS (recall Brier score decomposition in refinement and calibration), b) there is a potential excess risk due to ECE exceeding the ACCE. At this moment, no attack method is known that induces significant excess risk, but it is possible.
>
> We further discuss motivations to care about the ACCE in Section 2.2.
>
> To the best of our knowledge there does not exist a single method providing meaningful, *exact* guarantees during inference (including  certified accuracy) and hence all are an estimate.
>
> In the vast majority of applications, we do not forsee a dichotomy of exact certificate or no certificate. While such might exist, in most cases protecting the CBS exactly and estimating the CCE through ACCE (given how strong it is in comparison) will be *significantly* more informative than not doing this. We hope the reviewer agrees with us on this.

---

> > ### Author Response · Authors · 2024-12-04
> >
> > ### Literature
> > - Kumar, A., Sarawagi, S., & Jain, U. (2018). Trainable calibration measures for neural networks from kernel mean embeddings. In J. Dy & A. Krause (Eds.), *Proceedings of the 35th International Conference on Machine Learning* (Vol. 80, pp. 2805–2814). PMLR.
> > - Popordanoska, T., Sayer, R., & Blaschko, M. B. (2022). A consistent and differentiable Lp canonical calibration error estimator. In A. H. Oh, A. Agarwal, D. Belgrave, & K. Cho (Eds.), *Advances in Neural Information Processing Systems*.
> > - Bohdal, O., Yang, Y., & Hospedales, T. (2023). Meta-calibration: Learning of model calibration using differentiable expected calibration error. _Transactions on Machine Learning Research_.
> > - Murphy, A. H. (1973). A new vector partition of the probability score. _Journal of Applied Meteorology, 12_(4), 595–600.

---

### Official Review · Reviewer_2ZV2 · 2024-10-30

**Soundness:** 4
**Presentation:** 4
**Contribution:** 4
**Rating:** 8
**Confidence:** 4

**Summary:**

This paper discusses the importance of model calibration and solutions for determining a model's certified calibration and training models to be robust against calibration attacks. The paper starts by highlighting the importance of considering model calibration along with model accuracy by showing attacks degrading calibration and discussing applications in which this could have a strong negative consequences. The paper generalizes existing attacks on confidence estimation to form their new $(\eta, \omega)$-ACE attacks. The authors go on to describe certified Brier scores and certified calibration error providing a closed form solution for the first and detailing how to approximate the later with a MIP and solving it using ADMM. The paper then explores a new method of adversarial training building off SmoothAdv which they call adversarial calibration training (ACT). Finally, through an extensive set of experiments they show the effectiveness of ACT.

**Strengths:**

I found this paper to very well written and argued. The paper is structured well and there is an appropriate level of formality and theorems which motivate the work. To the best of my knowledge they propose the novel metrics of certified brier score and certified calibration error and give algorithms to compute these. They also provide a novel algorithm, ACT, for training models to have better uncertainty calibration. I also found the paper to have extensive experiments showing the performance of ACT in various scenarios.

**Weaknesses:**

1. The main issue I have with this paper is that it does not at all address the extensive work in non-probabilistic verification/certification of Neural Networks (Abstract Interpretation: IBP, CROWN, DeepPoly, etc. SMT Solvers: Ehlers et al 2017, Marabou, etc. Relaxation: Neurify, etc. Certified Training: IBP, DiffAI, COLT, CROWN, SABR, etc.). The paper defines formal certificates (C1, C2) but does not note that the certificates it proves/approximates are based on randomized smoothing and thus probabilistic in nature (i.e. certifying a score above a certain probability threshold). I am also curious as to what the authors think about using non-probabilistic verification/certified training methods in the calibration certification case. Smoothed classifiers tend to have good performance at the cost of inference time (i.e. performing 10s-100s of inference passes to get a classification). Do the authors think that adapting abstract interpretation based approaches could work in this case for verification or training?
2. This paper also lists training times in the 100s of GPU hours and verification times in the 1000s of GPU hours (Appendix N) using a total of 5000 GPU hours for the experiments in the paper (although admittedly there are many experiments). These finetuning times are approaching multibillion parameter LLM finetuning times.
3. While the authors bring up valid motivations in Section 2.2 I find them a bit hard to see in practice. Currently accuracy still dominates as a metric for most practical applications, most SOTA defense methods protecting model accuracy against attackers actually sacrifice accuracy and thus even these have a hard time finding real-world use. The premise of this paper leans on these methods being used but not being protective enough of calibration (although the papers experiments do show that their method can be used without sacrificing much certified accuracy). This point along with (2) make me think this method seems a bit hard to justify in practice.

**Minor Weaknesses**

1. While I found the paper to be well written and easy to follow, I think it would greatly benefit from some overarching example which could help provide insight and make Section 3 a bit easier to follow.

**Questions:**

The main reasoning for my score is the 3 points above in the weaknesses section, I am happy to raise my score if these are addressed. The following list of comments I think could make the paper clearer but I don't think significantly impact my grading.

**Clarity**

1. (line 160) When $\omega = \hat{y}$ it seems that Equation 2 would read as maximizing $\mathcal{L}_{CE}(f(x,\gamma), f(x,\gamma))$, I think the notation here is a bit unclear, I think you mean $\hat{y} = f(x)$?
2. (table 1) What metric is actually given here? The drop in AdaECE compared to the untracked version? I think at this point it is still slightly confusing how one should interpret these numbers, is higher better? worse? does it depend on $\eta$ and/or $\omega$?
3. (line 168) Can you be a bit clearer about how this adversarial training is happening? The existing model is adversarially finetuned? A new model is trained with adversarial training? Standard PGD adversarial training? Does the accuracy of the adversarially trained model change? Or reference somewhere where this information can be found.
4. (line 180) This is slightly confusing, a practitioner deploys a model with a threat model budget? I guess you are trying to say that the deployed model is certified for all $R \leq R_n$? How are they receiving a guarantee? What is being guaranteed? This should be individual certificates on individual points, is this being generalized to a data distribution? I think there should be a clearer way to defined the certified accuracy here.
5. (line 190) I think it is important to note that Gaussian smoothing gives probabilistic certificates. The certificate defined in C1,C2 are deterministic. There also exist methods for obtaining deterministic $l_p$ certificates through abstract interpretation/LP solvers/etc, so I think a distinction should be made here as these methods more accurately fit the certificates above.
6. I think throughout the Section 3 it would make the argument easier to follow if there was a toy example of some sort.

**Minor Comments**

1. (line 34) This sentence reads strangely maybe "...neural network classifiers **can be** extremely sensitive... human eye **for which** $x + \gamma$ ..."
2. (line 42) "and thus **provide guarantees on** model accuracy"? otherwise the last clause it strange
3. (line 45) "average mismatch"? undefined at this point
4. (line 128) extra comma
5. (line 137) There is no example being extended, it was mentioned in line 124 but has not been explained yet
6. (line 166) This is an incomplete sentence, "**For** a ResNet50 *model* on ImageNet, **a** $(1,y)$-ACE **attack**...$?
7. (line 168) missing a word after the comma "**but**" or "**however**", also should it be "a $(1,\hat{y})$" or "an $(1,\hat{y})$"
8. (line 184) I think the notation/wording is a bit confusing here. "**for all** perturbations **s.t.**"? I guess $\forall ||\gamma_n|| \leq R_n$ seems slightly weird I understand what is trying to be said $\forall \gamma_n \in \[\gamma \text{ s.t. } ||\gamma|| \leq R_n\]. F(x_n + y_n) = F(x_n)$. I guess you don't need to change this but it seems like a slight abuse of notation.
9. (line 184) Why is this an assumption? Isn't it more like a definition?
10. (line 208) Is this an assumption or are you saying that a model has these properties by definition, or clarify what the assumption is here i.e. is there some case where this doesn't hold?
11. (line 214) Maybe place a dot between $l$ and $c$ otherwise it kind of seems like $lc$ is a new variable.
12. (line 233) "that **is** possible"
13. (line 244) is this backward? should it be (4) to (5)?
14. (line 268) define $\mathbf{1}_M^\top$

---

> ### Author Response · Authors · 2024-11-23
> **Rebuttal 1/2**
>
> We thank the reviewer for taking the time to read our paper and the incredibly detailed feedback. We appreciate the reviewer agrees that our proposed metrics and training methods are "novel" and that our experiments are extensive and show the effectiveness of ACT. We are pleased they find our work "very well written and argued". We thank for the number of minor comments. We will not respond individually, but appreciate the feedback and incorporate the remarks into the paper.
>
> > W1a: [...] this paper [...] does not at all address the extensive work in non-probabilistic verification/certification of Neural Networks [...]
>
> As our method is agnostic towards *how* certificates **C1** and **C2** are obtained, a comparative study of verification techniques remains out-of-scope for this work. We utilize Gaussian Smoothing because it is popular and (as the reviewer notes) performs well. Most importantly, it scales well to modern sized neural networks (at the expense of inference cost). To the best of our knowledge none of the mentioned verification methods scale comparably.
>
> However, we agree with the reviewer that these methods should not remain unmentioned and now mention them in Appendix A.1 in which we provide a detailed background on certification and Gaussian Smoothing.
>
> > W1b: I am also curious as to what the authors think about using non-probabilistic verification/certified training methods in the calibration certification case. [...] Do the authors think that adapting abstract interpretation based approaches could work in this case for verification or training?
>
> We thank the reviewer for this interesting question. Yes, certified calibration is compatible with non-probabilistic / abstract interpretation methods as long as certificate **C1** and **C2** can be established. We also think that our ACT training can be combined with other certified training objectives (e.g. see Appendix J.4.2. L1869 we propose an alternative objective for ACT using the ACCE as penalty when updating weights). We leave a closer investigation of this for future work.
>
> As mentioned above, we use Gaussian smoothing for scalability. However, when using smaller networks, deterministic certification methods (e.g. IBP) can be superior and would likely be preferred in combination with certified calibration.
>
> > W1c: The paper defines formal certificates (C1, C2) but does not note that the certificates it proves/approximates are based on randomized smoothing and thus probabilistic in nature (i.e. certifying a score above a certain probability threshold).
>
> We agree with the reviewer that randomized smoothing practically is probabilistic. With small $\alpha$, however, the behavior approximates a deterministic certificate.
>
> Due to space constraints we omit this in the first part of Section 3. However, we do note this in Appendix A.1 (L796-L802) where we provide a more detailed account of Gaussian smoothing.
>
> > W2: [...] a total of 5000 GPU hours for the experiments in the paper (although admittedly there are many experiments). [...]
>
> The reported GPU hours are not indicative of the computational cost that practitioners would face applying our method. As the reviewer noted, our experiments are extensive across multiple datasets, hyperparameters and repetitions; they cover pre-training, fine-tuning, Gaussian Smoothing and our certification. The GPU hours refer to the cost of replicating our *entire* experiments. Practitioners can certify calibration error in <1 GPU hour and perform fine-tuning in low 10s of GPU hours. Certifying the Brier score in closed form has (almost) zero cost.
>
> > W3a: Currently accuracy still dominates as a metric for most practical applications
>
> Accuracy, while important, does not inform calibration (see Appendix B.1 for an explanation) and hence there is a need for both, accuracy and calibration. There are many applications where calibrated uncertainty quantification is very important (see surveys e.g. Wang 2023, Abdar 2021, Gawlikowski 2023). We believe, our work sheds some light on the over-reliance on accuracy in the robustness literature.

---

> ### Author Response · Authors · 2024-11-23
> **Rebuttal 2/2**
>
> > W3c: [...] most SOTA defense methods protecting model accuracy against attackers actually sacrifice accuracy and thus even these have a hard time finding real-world use. The premise of this paper leans on these methods being used but not being protective enough of calibration (although the papers experiments do show that their method can be used without sacrificing much certified accuracy).
>
> About general robustness literature: We agree with the reviewer that a significant portion of robustness and certification literature has trade-offs on the accuracy that are unfavorable, thus reducing their deployment in the real world. We still believe, however, that this research has merits and that there are a wide-range of security-critical applications where such trade-offs can be justified.
>
> As to our paper: If a system uses uncertainty estimates, estimating its calibration is vital. This incurs no cost towards accuracy. Hence, estimating the ACCE and CBS should not be considered part of a trade-off. For the adversarial calibration training (ACT), it is true that such trade-off can occur and that is a weakness in which our work follows existing work. However, as the reviewer notes, we find that this trade-off is very weak and accuracy can be retained when using our methodology. Hence, we think we make a compelling offer with ACT.
>
>
> ## Questions
>
> We incorporate all of our discussions here into the revised paper.
>
> > Q1: (line 160) When $\omega = \hat{y}$ it seems that Equation 2 would read as maximizing $\mathcal{L}_{CE}(f(x,\gamma), f(x,\gamma))$, I think the notation here is a bit unclear, I think you mean $\hat{y} = f(x)$?
>
> There was indeed an error in notation. We define $\hat{y} \triangleq F(x)$. E.g. when $\eta=1$, we maximize the loss w.r.t. the true label $y$ or the predicted label $\hat{y}$. We updated this in the revised paper.
>
> > Q2: (table 1) What metric is actually given here? The drop in AdaECE compared to the untracked version? I think at this point it is still slightly confusing how one should interpret these numbers, is higher better? worse? does it depend on $\eta$ and/or $\omega$?
>
> For the AdaECE and ECE, lower is better (here, we use a $ECE \in [0, 100]$ notation rather than [0, 1]). The table shows absolute AdaECE values. For instance, for ImageNet the calibation error (AdaECE) is 3.70 without attack on a vanilla model ("ST"=Standard Training). Using all combinations of $\eta$ and $\omega$ in Eq 2, yields calibration errors that range from AdaECE=1.06 (actually the attacked *improved* calibration) to AdaECE=47.23 (a 12-fold worsening of the calibration error).
>
>
> > Q3: (line 168) Can you be a bit clearer about how this adversarial training is happening? The existing model is adversarially finetuned? A new model is trained with adversarial training? Standard PGD adversarial training? Does the accuracy of the adversarially trained model change? Or reference somewhere where this information can be found.
>
> We believe this question is about the methodology of the motivating example. We fit the model using standard PGD training that solves the classical min-max-problem over the cross-entropy. We added a more detailed overview of the experimental setup to Appendix B.2.
>
> > Q5: (line 190) I think it is important to note that Gaussian smoothing gives probabilistic certificates [...]
>
> See our response to W1c.
>
> > Q6: I think throughout the Section 3 it would make the argument easier to follow if there was a toy example of some sort.
>
> We thank the reviewer for this great suggestion and agree that this would be helpful. The revised version now includes a working example in Appendix E.2.
>
> ## Literature
>
> * Wang, C. (2024). Calibration in deep learning: A survey of the state-of-the-art. arXiv.
> * Gawlikowski, J., Tassi, C.R.N., Ali, M. et al. A survey of uncertainty in deep neural networks. Artif Intell Rev 56 (Suppl 1), 1513–1589 (2023).
> * Moloud Abdar, Farhad Pourpanah, Sadiq Hussain, Dana Rezazadegan, Li Liu, Mohammad Ghavamzadeh, Paul Fieguth, Xiaochun Cao, Abbas Khosravi, U. Rajendra Acharya, Vladimir Makarenkov, Saeid Nahavandi,
> A review of uncertainty quantification in deep learning: Techniques, applications and challenges, Information Fusion,
> Volume 76, 2021

---

> > ### Comment · Reviewer_2ZV2 · 2024-11-24
> >
> > I am fairly satisfied with the changes/responses to my questions so I am increasing my score to an 8. I still have a little trouble with the argument that randomized smoothing scales well to modern sized neural networks. In this paper each inference requires 100k individual DNN executions, I don't see any modern application where this (or even a smaller amount say 100 executions) could be tolerated. Anyways, this is a general issue with randomized smoothing which is not specific to this paper so it does not affect my score.
> >
> > I also don't really like the idea that "With small $\alpha$, however, the behavior approximates a deterministic certificate." The aforementioned deterministic methods provide a proof that a network cannot be broken, with randomized smoothing unless you sample every single point (computationally infeasible even for small networks) you will never get a deterministic certificate. As our community is posing these methods for safety critical applications such as self driving or medicine there is an important distinction between a proof that a network cannot be attacked and an extremely high probability a network cannot be attacked.

---

> > > ### Author Response · Authors · 2024-11-28
> > >
> > > We thank the reviewer for their efforts, their insightful comments on certification and Gaussian smoothing, as well as for raising their score.
> > >
> > > If there are any further questions or points for clarification, we would be happy to address them.

---

### Official Review · Reviewer_Xy4o · 2024-11-01

**Soundness:** 3
**Presentation:** 3
**Contribution:** 3
**Rating:** 8
**Confidence:** 4

**Summary:**

In this paper, the authors show that it is possible to generate adversarial examples that don't change accuracy but impact confidence scores heavily. Thus, confidence scores can not really be trusted by introducing a parametrized attack that can raise and lower confidence scores without impacting the classification. Then they introduce *certified calibration*, a method to provide guarantees on the calibration of certified models under adversarial attacks. Then a training scheme is proposed to improve the certified uncertainty calibration.

**Strengths:**

The certified calibration is an important and currently not very developed problem. The authors do a good job in motivating the problem and proposing solutions. The paper is mostly easy to follow, though the reviewer feels that illustrations would sometimes have improved readability. The approach is novel and interesting.

**Weaknesses:**

- It is unclear how large the effect is in Table 2. The Brier Score although defined in simple terms is appears somewhat uninterpretable.
- The authors did not report mean and std of their results over different seeds.
- The reviewer has the feeling that explanations could be made more concise from time to time while improving readability further.

Minor:
- 232/233 - the sentence here does not appear to be grammatically correct

**Questions:**

- For Table 1: could the authors (possibly via bootstrapping) estimate variances here?
- What is the final loss that is optimized to solve equation 2? As in how did you encode it?
- Is the certificate actually sound or just approximate (see Line 238)?
- Is $a_{i,j}$ only for 1 $j$ allowed to be 1 while the others are 0?
- What where the considerations for choosing ADMM?
- Does the reviewer suppose correctly that the runtimes in G1 where reported for CIFAR-10?

---

> ### Author Response · Authors · 2024-11-22
> **Rebuttal**
>
> We thank the reviewer for their efforts to review our work. We appreciate the reviewer agrees that certified calibration is "important", understudied, and our methods are "novel". We are pleased the reviewer finds the paper "easy to follow".
>
> > W1: It is unclear how large the effect is in Table 2. The Brier Score although defined in simple terms is appears somewhat uninterpretable.
>
> Could the reviewer please clarify what they find confusing. We are more than happy to clarify and update the table, but are uncertain what the reviewer is referring to.
>
> > W2: The authors did not report mean and std of their results over different seeds.
>
> This is indeed correct. Generally, in this work we estimate worst-cases and hence $mean$ aggregation is often less appropriate than $max$ aggregation. Where possible, we do provide insights into variation through replications. Two examples:
>
> - Figure 3: We replicate Figure 3 (showing the ACCE computed through ADMM vs dECE) in Figures 13 and 14. Specifically, in Figure 13 we show the *maximum* achieved by any numerical method (i.e. the closest approximation to the worst case achievable). This is done over 1000s of initializations. In Figure 14, we show that *one* set of hyperparameters for ADMM outperforms the $max$ over 1000s of hyperparameter combinations for the dECE.
>
> - Table 2: This table shows the minimum calibration error across many fine-tuned models with accuracies within a certain range. In addition, we present results from a slightly different angle, actually plotting *raw* data  in Figure 4.
>
> We hope the reviewer agrees that we provide insights into the reliability of our results. If we missed the reviewers point, could the reviewer please be more specific and we will gladly consider making adaptations.
>
> > W3: The reviewer has the feeling that explanations could be made more concise from time to time while improving readability further.
>
> This work requires previous knowledge on uncertainty calibration and certification, and hence we anticipate many readers might not be familiar with one or the other. We agree that this paper could be more concise, but in the interest of helping those readers we opted for a slightly slower approach.
>
> > Q1: For Table 1: could the authors (possibly via bootstrapping) estimate variances here?
>
> We agree with the reviewer that variances of the attack results could be estimated. However, as we do not make comparative claims on the magnitude of the effect (e.g. we do not claim SOTA in Table 1), we opted to invest our compute in other parts of the paper. We believe the results in Table 1 in their form sufficiently motivate the paper by showing that the calibration is vulnerable.
>
> > Q2: What is the final loss that is optimized to solve equation 2? As in how did you encode it?
>
> We use $\eta \mathcal{L}_{CE}(f(x+\gamma),\omega)$ (see Eq 2) directly as loss. We perform projected gradient descent to satisfy $\|\gamma\|\leq \epsilon$ and stop optimisation if the next step would violate $F(x+\gamma)=F(x)$. As described in our answer to Q1, the purpose of this attack is to show how trivially it is possible to alter calibration, not establishing any SOTA. Hence, we focused our attention on our own methods rather than iterating to improve the attack in Eq 2.
>
> > Q3: Is the certificate actually sound or just approximate (see Line 238)?
>
> The certified calibration error (CCE, Eq 4) by definition is the true upper bound. The exact solution to the MIP in Eq 6 is also a true upper bound. The approximate calibration error (ACCE) is a numerical approximation. Obtaining an exact CCE is challenging due to the discrete, non-convex nature of the problem. Finding non-vacuous upper bounds is also challenging (e.g. tools such as Minkowski inequality yield vacuous bounds). The certified Brier score (CBS) is a true upper bound and tight.
>
> > Q4: Is $a_{i,j}$ only for 1 $j$ allowed to be 1 while the others are 0?
>
> Yes, that is correct and enforced through the "Unique Assignment Constraint" (see L279 to L283).
>
> > Q5: What where the considerations for choosing ADMM?
>
> The optimization problem is mixed-integer and non-convex, for which no "canonical" optimizer exists to the best of our knowledge. ADMM has been shown to posses good convergence on other non-convex problems (see L314, Wang, 2019), and is simple to implement using software packages such as PyTorch or JAX. After initial experiments, we found it easy to get consistent results with fairly standardized hyperparameters. In addition, it is very fast and scalable.
>
> > Q6: Does the reviewer suppose correctly that the runtimes in G1 were reported for CIFAR-10?
>
> Yes, that is correct. We updated G.1 to clarify.

---

> > ### Author Response · Authors · 2024-11-22
> > **Literature**
> >
> > Here we provide the literature referenced in the rebuttal above.
> >
> > * Yu Wang, Wotao Yin, and Jinshan Zeng. Global convergence of ADMM in nonconvex nonsmooth optimization. Journal of Scientific Computing, 78:29–63, 2019.
> > * Hadi Salman, Jerry Li, Ilya Razenshteyn, Pengchuan Zhang, Huan Zhang, Sebastien Bubeck, and Greg Yang. Provably Robust Deep Learning via Adversarially Trained Smoothed Classifiers. Neurips 2019.
> > * Jeremy Cohen, Elan Rosenfeld, and Zico Kolter. Certified Adversarial Robustness via Randomized Smoothing, ICLR 2019.

---

> > > ### Comment · Reviewer_Xy4o · 2024-12-01
> > >
> > > I thank the authors for their reply. I will follow up on the questions the authors raised:
> > >
> > > **Could the reviewer please clarify what they find confusing. We are more than happy to clarify and update the table, but are uncertain what the reviewer is referring to.**
> > >
> > > It is unclear to me what the effect of an ACCE of 49.30 is vs an ACCE 41.32 (Table 3, certified radius = 0.5). While i do understand that lower is better, what is the effect of this difference of roughly 8? Is it a large difference? A small difference? For example (although not applicable here directly), accuracy for a classification task is very interpretable as 8 more precentage points in accuracy mean that in expectation 8 more inputs are classified correctly.
> > >
> > > **If we missed the reviewers point, could the reviewer please be more specific and we will gladly consider making adaptations.**
> > >
> > > My question is somewhat related to the previous point. For example in Figure 3, it is unclear if ADMM, dECE and Brier are successively worse by chance or if the improvement is statistically significant. It also would reveal information about the stability of the method to hyper parameters. The Figure 13 and 14 provide some value in that regard.

---

> > > > ### Author Response · Authors · 2024-12-04
> > > >
> > > > We thank the reviewer for their continued engagement in the review process, their clarifications and gladly attempt to clarify from our side.
> > > >
> > > > > It is unclear to me what the effect of an ACCE of 49.30 is vs an ACCE 41.32 (Table 3, certified radius = 0.5). While i do understand that lower is better, what is the effect of this difference of roughly 8? Is it a large difference? A small difference? For example (although not applicable here directly), accuracy for a classification task is very interpretable as 8 more percentage points in accuracy mean that in expectation 8 more inputs are classified correctly.
> > > >
> > > > At convergence the ACCE is an exact expected calibration error (ECE), that means we can interpret the ACCE as such. For example, consider $ECE=0.493$ (let's use a $[0,1]$ scaling here): For a datapoint with confidence $z$, we expect the probability of being correct (i.e. $P(y=\hat{y}|z)$) to be 0.493 away from the confidence $z$. An improvement to $ECE=0.4132$  ($\Delta \approx 0.08$ as noted by the reviewer), means we now expect the confidence score to be 8% closer to the true probability of this sample being classified correctly. Returning to the ACCE, instead of indicating an observed improvement of 8%, it indicates that under attack the worst we could observe is now improved by 8%.
> > > >
> > > > Judging the practical impact of an $ECE=0.49$ and an improvement of $0.08$ is not trivial and depends on the context. Generally, on clean data researchers consider $ECE>0.05$ to be poor (very roughly). However, under distribution shift or attack (e.g. see Table 1) calibration deteriorates rapidly. Ovadia et al (2019) provide a discussion of calibration under distribution shift. They report calibration errors up to $0.63$ on CIFAR-10. They compare various methods for uncertainty quantification and regard differences between methods of around $0.05$ to $0.1$ as strong. This gives us confidence to state that the improvements we achieve are substantial.
> > > >
> > > > > My question is somewhat related to the previous point. For example in Figure 3, it is unclear if ADMM, dECE and Brier are successively worse by chance or if the improvement is statistically significant. It also would reveal information about the stability of the method to hyper parameters. The Figure 13 and 14 provide some value in that regard.
> > > >
> > > > We are glad Figures 13 and 14 could provide valuable insights.
> > > >
> > > > Ovadia, Y., Fertig, E., Ren, J., Nado, Z., Sculley, D., Nowozin, S., Dillon, J.V., Lakshminarayanan, B., & Snoek, J. (2019). Can You Trust Your Model's Uncertainty? Evaluating Predictive Uncertainty Under Dataset Shift. _Neural Information Processing Systems_.

---

### Official Review · Reviewer_apBk · 2024-11-04

**Soundness:** 3
**Presentation:** 3
**Contribution:** 3
**Rating:** 6
**Confidence:** 3

**Summary:**

This work addresses a critical vulnerability in neural network classifiers by examining how adversarial attacks can significantly harm model calibration (the reliability of confidence scores) while leaving accuracy unchanged - a particular concern for safety-critical applications. The authors introduce two new metrics for certified calibration: the Certified Brier Score (CBS) with a closed-form bound, and the Certified Calibration Error (CCE) approximated through mixed-integer programming. They also propose a novel training method called Adversarial Calibration Training (ACT) with two variants (Brier-ACT and ACCE-ACT) to improve certified calibration while maintaining accuracy.

**Strengths:**

1. Identifies an important security vulnerability (calibration under adversarial attacks) that wasn't previously well-studied
2. Well-formulated mathematical foundations for certified calibration

**Weaknesses:**

1. The entire certification framework relies heavily on the binning strategy used in ECE computation
2. Complex hyperparameter tuning required for ADMM solver

**Questions:**

1. How does the certification process scale with the number of classes? Would the approach be practical for problems with hundreds or thousands of classes?
2. Could the proposed methods be extended to other uncertainty estimation approaches beyond softmax confidence scores?
3. How sensitive is the certification to the choice of binning strategy in the ECE computation?

Additional Advice:
You should follow the ICLR submission format.

---

> ### Author Response · Authors · 2024-11-22
> **Rebuttal**
>
> We thank the reviewer for their efforts and recognizing the importance of our work in identifying a "critical vulnerability in NN classifiers" and the "well-formulated mathematical foundation" of certified calibration.
>
> > W1: The entire certification framework relies heavily on the binning strategy used in ECE computation
> > Q3: How sensitive is the certification to the choice of binning strategy in the ECE computation?
>
> We do not see a strong impact of the binning scheme on the ACCE. The ACCE does indeed depend on the binning as we use the canonical estimator (see Eq 1). However, we find strong correlations between certified Brier score (CBS) and the ACCE (r=0.98) (see L494-495). The CBS does *not* depend on any binning and is available in closed-form, suggesting that binning does not confound the ACCE significantly.
>
> > W2: Complex hyperparameter tuning required for ADMM solver
>
> While the reviewer is correct in noting the large number of hyperparameters required for ADMM, we have performed extensive experimentation and identified a set of hyperparameters for the ADMM solver that works very well across all our experiments. We list these parameters in Appendix I.1. Using any of these or the ensemble (as we do in Section 5.3) will likely yield good convergence characteristics.
>
> Additionally, in Figure 14 we show that a single set of hyperparameters for the ADMM solver (almost uniformly) outperforms the maximum over 1000s of hyperparameter sets for the dECE across datasets and certified radii. We strongly believe the evidence provided suggests that the hyperparameters we present can be expected to work well in a very wide range of settings.
>
> The certified Brier score is available in closed form and does not require any hyperparameters.
>
>
> > Q1: How does the certification process scale with the number of classes? Would the approach be practical for problems with hundreds or thousands of classes?
>
> The answer depends on the metric:
>
> CBS: The Brier score itself depends on the number of classes $K$ linearly, $\mathcal{O}(K)$. The CBS inherits that.
>
> ACCE: The certification only depends on the correctness of each prediction. Once that is established, obtaining the ACCE has constant complexity in the number of classes $\mathcal{O}(1)$.
>
> Hence, we propose that our method scales favourably with large number of classes, far beyond the 1K classes in our ImageNet experiments.
>
> > Q2: Could the proposed methods be extended to other uncertainty estimation approaches beyond softmax confidence scores?
>
> Yes, our method is agnostic to uncertainty estimation techniques. All that matters is that the certificates **C1** and **C2** can be established. For instance, Gaussian smoothing can be performed with temperature scaling or over the composition of neural network + Platt scaling establishing **C1** and **C2** without any adaptations (i.e. Lemma 2 in Salman et al. (2019) still holds when the smoothed function is the composition of neural network + confidence postprocessing).
>
> #### Literature
> * Hadi Salman, Jerry Li, Ilya Razenshteyn, Pengchuan Zhang, Huan Zhang, Sebastien Bubeck, and Greg Yang. Provably Robust Deep Learning via Adversarially Trained Smoothed Classifiers. Neurips 2019.

---

> > ### Comment · Reviewer_apBk · 2024-12-02
> >
> > Having reviewed the authors' responses regarding binning strategy and hyperparameter tuning, I appreciate their thorough clarifications and updates. I also note that Reviewer 2ZV2 found the mathematical foundations solid and gave a strong positive recommendation, while Reviewer oWYq raised important concerns about approximation certification. After considering all viewpoints and the authors' detailed responses, I am updating my score from 5 to 6.

---

> > > ### Author Response · Authors · 2024-12-04
> > >
> > > We thank the reviewer for their efforts to read through rebuttals and discussions. We are glad these could address the reviewer's concerns and thank the reviewer for raising their score.

---

### Meta-Review · Area_Chair_CQuc · 2024-12-21

**Metareview:**

This paper studies a seemingly new and interesting problem: certifying uncertainty calibration of predictive models under adversarial attacks. The authors demonstrate that attacks can be devised that drastically change the calibration of preditors without necessarily changing the predicted label, thus still constituting a threat. The paper introduces two metrics for certified calibrations (a certified Brier Score, with closed-form bound, and a certified calibration error, approximation through a mixed-integer program). The paper also proposes to devise models that are robust to these attacks through a calibration version of adversarial training, with two variants.

**Strengths**
* The paper studies an important and overlooked problem in adversarial robustness.
* The paper is clearly written and easy to read.
* The solutions are novel and interesting.

**Weaknesses**
After the exchanges with reviewers, the main weakness that I recognize is the fact that only an approximation to the certified calibration error (the ACCE).

**Summary**

The vast majority of the comments from the reviewers pertained to clarifications on the experimental results, which were all addressed. The main caveat was raised by Rev oWYq, who questions the utility and correctness of studying certificates that are only *approximate*, like the one developed here. This led to a productive exchange with the authors, which resulted in both agreeing to disagree. I regard these discussions as a healthy conversation on a good paper, and while I agree with Rev oWYq in the limited serious utility of the ACCE, I also concur with the authors that useful insights can be derived from it (such as the adversarial training procedure that they present). For these reasons, I'm recommending acceptance.

**Additional Comments On Reviewer Discussion:**

Exchanges between the authors and the reviewers were productive, and resulting in the clarification of experimental details, typos, and small mistakes in presentation. The central argument concerned the limitation of an approximate certificate (ACCE), as raised by Rev oWYq. This led to an insightful discussion, reflecting different perspectives that, I believe, coexist in the community of this conference (see above).

---

### Decision · Program_Chairs · 2025-01-22

Accept (Poster)